



# A fine-scale digital elevation model of Antarctica derived from ICESat-2

Xiaoyi Shen[1,2], Chang-Qing Ke[1,2], Yubin Fan[1,2], Lhakpa Drolma[3]

[1] School of Geography and Ocean Science, Nanjing University, Nanjing, 210023, China

[2] Jiangsu Provincial Key Laboratory of Geographic Information Science and Technology, Nanjing University, Nanjing, 210023, China

[3] Institute of Tibetan Plateau Atmospheric and Environmental Sciences, Tibet Meteorological Bureau, Lhasa, 850000, China

*Correspondence to*: Chang-Qing Ke (kecq@nju.edu.cn)

**Abstract.** Antarctic digital elevation models (DEMs) are essential for human fieldwork, ice topography monitoring and ice

mass change estimation. In the past thirty decades, several Antarctic DEMs derived from satellite data have been published. However, these DEMs either have coarse spatial resolutions or vague time stamps, which limit their further scientific applications. In this study, the new-generation satellite laser altimeter Ice, Cloud, And Land Elevation Satellite-2 (ICESat-2) is used to generate a fine-scale and specific time-stamped Antarctic DEM for both the ice sheet and ice shelves. Approximately $4.69 \times 10^9$ ICESat-2 measurement points from November 2018 to November 2019 are used to estimate

surface elevations at resolutions of 250 m, 500 m and 1 km based on a spatiotemporal fitting method, which results in a modal resolution of 250 m for this DEM. Approximately 74% of Antarctica is observed, and the remaining observation gaps are interpolated using the ordinary kriging method. National Aeronautics and Space Administration Operation IceBridge (OIB) airborne data are used to evaluate the generated Antarctic DEM (hereafter called the ICESat-2 DEM) in individual Antarctic regions and surface types. Overall, a median bias of 0.11 m and a root-mean-square deviation of 8.27 m result from

approximately $1.4 \times 10^5$ spatiotemporally matched grid cells. The accuracy and uncertainty of the ICESat-2 DEM vary in relation to the surface slope and roughness, and more reliable estimates are found in the flat ice sheet interior. The ICESat-2 DEM is superior to previous DEMs derived from satellite altimeters for both spatial resolution and elevation accuracy and comparable to those derived from stereo-photogrammetry and interferometry. The decimeter-scale accuracy and specific time stamp make the ICESat-2 DEM an essential addition to the existing Antarctic DEM groups, and it can be further used

for other scientific applications.

## 1 Introduction

Knowledge of the detailed surface topography in Antarctica is essential for human fieldwork, ice motion tracking and mass balance estimation (Sutterley et al., 2014; Bamber et al., 2009). Digital elevation models (DEMs) of Antarctica, for example, can be used for monitoring the topography of ice sheets and ice shelves and thus provide a crucial reference for ice dynamics



and glacier velocities (Slater et al., 2018; Wesche et al., 2007), which is necessary for Antarctic mass balance monitoring and potential sea level rise contribution estimation (Mengel et al., 2018; Ritz et al., 2015).

Due to the remoteness of Antarctica, most of the previously published Antarctic DEMs were derived from satellite or airborne data, e.g., elevation measurements from radar altimeters (Slater et al., 2018; Helm et al., 2014; Fricker et al., 2000), laser altimeters (DiMarzio et al., 2007), a combination of radar and laser altimeters (Bamber et al., 2009), stereo-
photogrammetry (Howat et al., 2019; Cook et al., 2012; Korona et al., 2009) and interferometry (Wessel et al., 2021). The currently available continent-scale Antarctic DEMs include one DEM derived from ICESat (hereafter called the ICESat DEM, DiMarzio et al., 2007), one based on the combination of ICESat and ERS-1 elevation measurements (hereafter called the ICESat/ERS-1 DEM, Bamber et al., 2009), two DEMs derived from CryoSat-2 (hereafter called the Helm CryoSat-2 DEM (Helm et al., 2014) and Slater CryoSat-2 DEM (Slater et al., 2018)), one DEM derived from stereo-photogrammetry
using GeoEye-1 and WorldView-1/2/3 imageries (hereafter called the Reference Elevation Model of Antarctica (REMA) DEM, Howat et al., 2019), and one DEM derived from Interferometric Synthetic Aperture Radar (InSAR) using TerraSAR-X and TanDEM-X data (hereafter called the TanDEM-X PolarDEM, Wessel et al., 2021).

All these DEMs provide reasonable elevation estimates for Antarctica; however, some flaws still cannot be totally avoided. The coverage of the ICESat DEM is limited in ice sheet margins due to its coarse across-track resolution (usually larger than
250 m). Although the ICESat/ERS-1 DEM improves the coverage by combining the measurements from ICESat and ERS-1 elevations, the specific time stamp of the DEM is still missing due to the different timespans (1994-1995 for ERS-1 and 2003-2008 for ICESat) of these two satellite altimeter datasets. This issue also exists with the REMA DEM and TanDEM-X PolarDEM, where multiyear satellite imageries were used. Different from the abovementioned DEMs, the Slater CryoSat-2 DEM was derived based on a model fitting method by using seven-year CryoSat-2 data (from July 2010 to July 2016). This
method can quantify the measured elevation fluctuations due to seasonal variations, and the time stamp is definitive. However, the penetration depth of the CryoSat-2 Ku-band into Antarctic dry snowpack is still unknown, which includes some uncertainties in the elevation estimation. A similar problem also exists with the Helm CryoSat-2 DEM and TanDEM-X PolarDEM (the penetration depth of the X-band into snow may be several meters, Fischer et al., 2020; Dehecq et al., 2016). A fine-scale Antarctic DEM with a definitive time stamp is still lacking.

The new-generation satellite laser altimeter Ice, Cloud, And Land Elevation Satellite-2 (ICESat-2) of the National Aeronautics and Space Administration (NASA), which was launched on 15 September 2018, provides near-global (up to 88°S) and dense land ice elevation measurements in an accurate repeated cycle of 91 days by using a multibeam (six beams in three pairs that work at 532 nm) laser altimeter (i.e., Advanced Topographic Laser Altimeter System, ATLAS, Neumann et al., 2019). The narrow footprint (approximately 17 m with a spatial interval of 0.7 m) and three pairs of beams (two beams
in one pair can determine the local slope) enable a fine-scale measurement of Antarctic surface heights even in steep regions. Hence, ICESat-2 can be expected to provide a new and specific time-stamped Antarctic DEM on a fine scale.



Here, we use a one-year time series (from November 2018 to November 2019) of ICESat-2 elevation measurements to generate a new Antarctic DEM that covers both the ice sheet and ice shelves (hereafter called the ICESat-2 DEM). The applied data, DEM generation method and quality control criteria are presented in Section 2. Furthermore, we present the

map of the ICESat-2 DEM and construct an accuracy evaluation by comparing it to the spatiotemporally matched elevation measurements from the NASA Operation IceBridge (OIB) airborne mission in Section 3. The performances of the ICESat-2 DEM and six currently available Antarctic DEMs are compared in Section 4, and Section 5 concludes this study.

## 2 Data and Methods

### 2.1 ICESat-2 data

The ICESat-2 ATL06 land ice elevation product (Smith et al., 2019) from November 2018 to November 2019 is used. This product provides land ice elevation measurements at a spatial resolution of 20 m after correcting instrument-specific biases (i.e., corrections for transmit-pulse shape and first-photon bias, Neumann et al. 2019); here, only ATL06 data with good quality (those for which atl06_quality_summary equals zero) are used to generate the DEM. For the data collected over Antarctic ice shelves, corrections for ocean tide and inverse barometer effects are also applied (Padman et al., 2002; Egbert

and Erofeeva, 2002, Egbert et al., 1994). Elevation measurements from all six beams are used to produce the densest surface height coverage. Although the signal energies of strong and weak beams are different, all six beams provide centimeter-scale elevation measurements, and the biases of two beams in one pair are less than 2 cm (Brunt et al., 2019) and 5 cm (Shen et al., 2021) for flat and steep surfaces. Thus, the effect of elevations estimated from weak beams can be negligible.

### 2.2 NASA OIB airborne data

Elevation measurements from the OIB airborne mission in Antarctica are used here to evaluate the accuracy of the ICESat-2 DEM on a continental scale, including in the stable ice sheet interior and active marginal ice shelves. Surface heights from OIB airborne missions are measured by the Airborne Topographic Mapper (ATM), a conically scanning laser altimeter (at 532 nm) with a swath width of 140 m and footprint size of 1 to 3 m. The elevation measurement accuracy of ATM is approximately 10 cm or better (Kurtz et al. 2013). Here, the IceBridge ATM L2 Icessn elevation, slope and roughness (V002)

product (Studinger et al., 2014) is used, and a data filter is applied to remove abnormal values due to geolocation errors or cloud cover. The local terrain parameters, i.e., slope and roughness, are calculated following Shen et al. (2021). To reduce the effect of seasonal elevation changes on DEM evaluation, the time difference between applied OIB airborne data and ICESat-2 DEM should be less than one year. Thus, OIB airborne data in October and November 2018 and October and November 2019 in Antarctica (Fig. 1) are chosen to evaluate the accuracy of the ICESat-2 DEM.




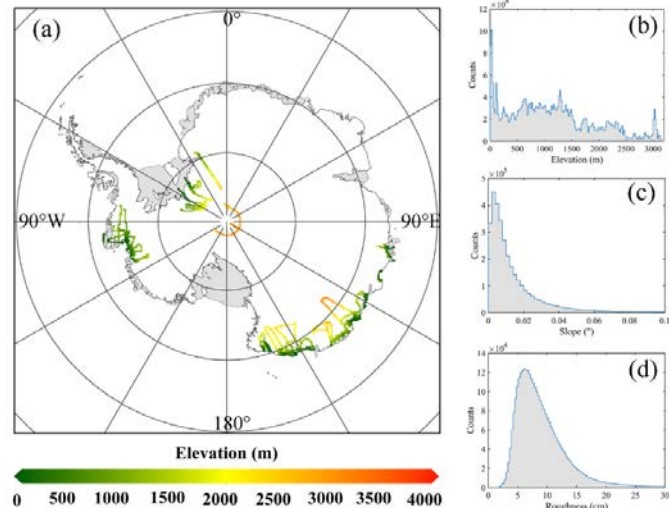

**Figure 1.** (a) Map of the applied OIB airborne data in October and November 2018 and October and November 2019 in Antarctica. This dataset covers both the flat ice sheet interior/ice shelves and steep ice sheet margins. Histograms of the OIB surface elevation, slope and roughness are shown in (b), (c) and (d), respectively.


## 2.3 Previously published Antarctic DEMs

Six previously published Antarctic DEM products are compared to the ICESat-2 DEM, i.e., ICESat DEM (DiMarzio et al., 2007), ICESat/ERS-1 DEM (Bamber et al., 2009), Helm CryoSat-2 DEM (Helm et al., 2014), Slater CryoSat-2 DEM (Slater et al., 2018), REMA DEM (Howat et al., 2019) and TanDEM-X PolarDEM (Wessel et al., 2021), as shown in Section 4.

Detailed information concerning these DEMs is provided in Table 1, and all DEMs have been referenced to the WGS84 ellipsoid.







**Table 1.** Detailed introductions to six previously published Antarctic DEMs, including the source data, time span of the source data, spatial resolution, time stamp and spatial coverage.

| DEM | Source data | Time span of applied source data | Spatial resolution | Time stamp | Coverage |
|---|---|---|---|---|---|
| ICESat DEM | ICESat | February 2003 to June 2005 | 500 m | Unclear | Ice sheet |
| ICESat/ERS-1 DEM | ICESat, ERS-1 | 1994-1995, 2003-2008 | 1 km | Unclear | Ice sheet |
| Slater CryoSat-2 DEM | CryoSat-2 | July 2010 to July 2016 | Variable resolutions,1, 2, and 5 km, modal resolution of 1 km | July 2013 | Pan-Antarctica |
| Helm CryoSat-2 DEM | CryoSat-2 | Jan 2011 to Jan 2014 | 1 km | Unclear | Pan-Antarctica |
| REMA DEM | GeoEye-1, WorldView-1/2/3 | 18 Sep 2007 to July 2017 | Variable resolutions,2 and 8 m | Unclear | Pan-Antarctica |
| TanDEM-X PolarDEM | TerraSAR-X, TanDEM-X | July 2016 to September 2017 | 90 m | Unclear | Pan-Antarctica |

## 2.4 ICESat-2 DEM generation method

### 2.4.1 Surface elevation and uncertainty estimation

To generate a definite time-stamped DEM and reduce the effect of seasonal elevation changes, following Slater et al. (2018), a model fitting method is applied. The elevation is estimated using a quadratic function based on the local surface terrain and a time term (Eq. 1). This function is fitted in each grid (the resolutions are listed in the following subsection) by using an iterative least-squares fit to all the elevation measurements. By considering the surface elevation fluctuations and seasonal changes, this method tends to obtain more accurate elevation estimates (McMillan et al., 2014; Flament et al., 2012).

$$E(x, y, t) = \overline{E} + a_0 x + a_1 y + a_2 x^2 + a_3 y^2 + a_4 xy + a_5 (t - t_{May\,2019}) \qquad (1)$$



Where $E$ is the surface elevations derived from ICESat-2 measurement points, $x$ and $y$ are the local surface terrain respectively, $t$ is the time term, and $\overline{E}$ is the DEM value in May 2019.

To reduce the effect of any poor fit, a quality control criterion listed in Table 2 is performed, which includes the number of ICESat-2 measurement points used, the time span of the data used, the root-mean-square deviation (RMSD) of the residuals of fitted elevations, the elevation rate of change and its uncertainty. These criteria are constructed for all grid cells, and thus, there are some elevation gasps in the initial DEM. The remaining gaps are filled by using ordinary kriging interpolation, which is widely used for generating previous DEMs (Helm et al., 2014; Slater et al., 2018). During the interpolation process,

a search radius of 10 km is applied to obtain neighboring measurement points. This elevation estimation model has been evaluated by previous studies (Slater et al., 2018; Konrad et al., 2017; McMillan et al, 2014; Moholdt et al., 2010; Smith et al., 2009), and the evaluation in Section 3.2 also demonstrates its validity.

**Table 2.** Quality control criteria applied to remove the unrealistic elevations due to the poor fitting performances in each grid
cell.

| Parameters | Rules |
| --- | --- |
| The number of ICESat-2 measurement points | $\leq 10$ |
| The time span | $\leq 2$ months |
| RMSD of the residuals of fitted elevations | $\geq 10$ m |
| Elevation change rate | $\geq 10$ m/yr |
| The uncertainty of elevation change rate | $\geq 10$ m/yr |

The performance of this surface fit method is also affected by the spatial distribution and number of ICESat-2 measurement points. After quality control, $4.69 \times 10^9$ ICESat-2 measurement points from November 2018 to November 2019 that cover all of Antarctica are used. An adequate number of ICESat-2 measurement points in one grid cell is required

to generate valid elevation estimates. Fig. 2 shows the distribution of the numbers of ICESat-2 measurement points used in individual grid cells (at a resolution of 1 km), which indicates a latitude-dependent pattern. Each grid cell contains approximately $418 \pm 310$ ICESat-2 measurement points. In the ice sheet interior, the large coverage of ICESat-2 measurement points provides a complete surface height observation. In the low-latitude region, the numbers of ICESat-2 measurement points are relatively small, the proportion of observed grid cells is reduced, and the representativeness is also

reduced.

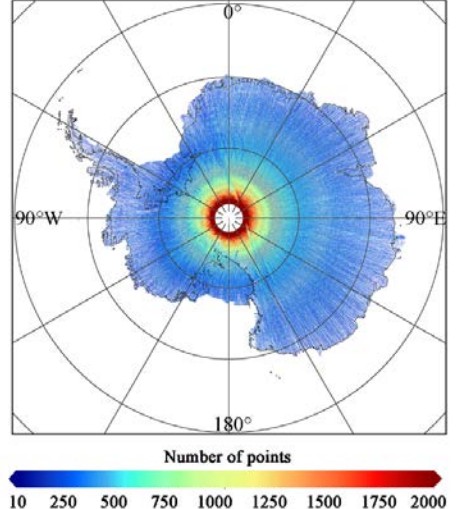

**Figure 2.** Map of the numbers of valid ICESat-2 measurement points in each 1 km grid cell. The numbers of ICESat-2 measurement points in grid cells at 250 m and 500 m are both resampled to a resolution of 1 km.


DEM uncertainties are calculated for observed and interpolated grid cells, respectively. The observed grid cell uncertainty is calculated based on the fitting performance, which is provided as the 95% confidence level for elevation estimation. For the interpolated grid cells, uncertainty is calculated from the kriging variance error. In the ICESat-2 DEM uncertainty calculation, the uncertainty from ICESat-2 measurements is not considered because the effect of ICESat-2 measurement bias is limited (< 5 cm, Brunt et al., 2019; < 14 cm, Shen et al., 2021) and no systemic error was found (Shen et al., 2021).

### 2.4.2 Choice of DEM resolution

The selection criterion of DEM resolution is to present the detailed pattern of elevations and ensure enough spatial coverage of observed elevations (a smaller resolution tends to cause more observed elevation gaps). Although a much finer scale (e.g., 250 m) can reveal a more detailed elevation pattern, this contributes to more gaps among observed elevations. To acquire the optimal compromise between the spatial resolution and spatial coverage of observed elevations, we calculate the variations in the spatial coverages of observed grid cells at different latitudes at variable spatial resolutions (250 m, 500 m and 1 km, which are usually applied in the Antarctic DEM, Fig. 3a). The overall spatial coverages of observed elevations when applying 250 m, 500 m and 1 km resolutions are 26%, 46% and 72%, respectively, and high-latitude areas always have higher observed elevation coverages. A single resolution cannot obtain ideal spatial coverage, especially in low-altitude areas. To increase the coverages of observed elevations as much as possible, referring to Slater et al. (2018), three spatial resolutions are used to estimate the surface elevations from ICESat-2. That is, elevations are estimated at resolutions of 250 m, 500 m and 1 km. The observation gaps in the 250 m DEM are filled by the resampled 500 m and 1 km DEMs (both resampled to the 250 m and 500 m DEM first). The addition of DEMs at 500 m and 1 km greatly increases the observation





coverage, the overall spatial coverage is approximately 74%, and the remaining gaps are filled using ordinary kriging

interpolation. Although three resolutions are applied, 500 m, 1 km and interpolated elevations are all resampled to a

resolution of 250 m to provide a consistent DEM dataset; hence, the final ICESat-2 DEM has a nominal resolution of 250 m.

The application of three resolutions may include additional effects, i.e., different grid cell resolutions tend to present

different elevation estimates. Here, we compare the elevation difference at the regional scale at different spatial resolutions.

The elevation values become lower when a larger spatial resolution is applied, which acts as a 'running mean' (Fig. 3b).

Although applying different spatial resolutions affects the elevation values, this method can increase the coverage of

observed elevations, and observed elevations tend to be more reliable than interpolated elevations (as shown in Section 3.2).

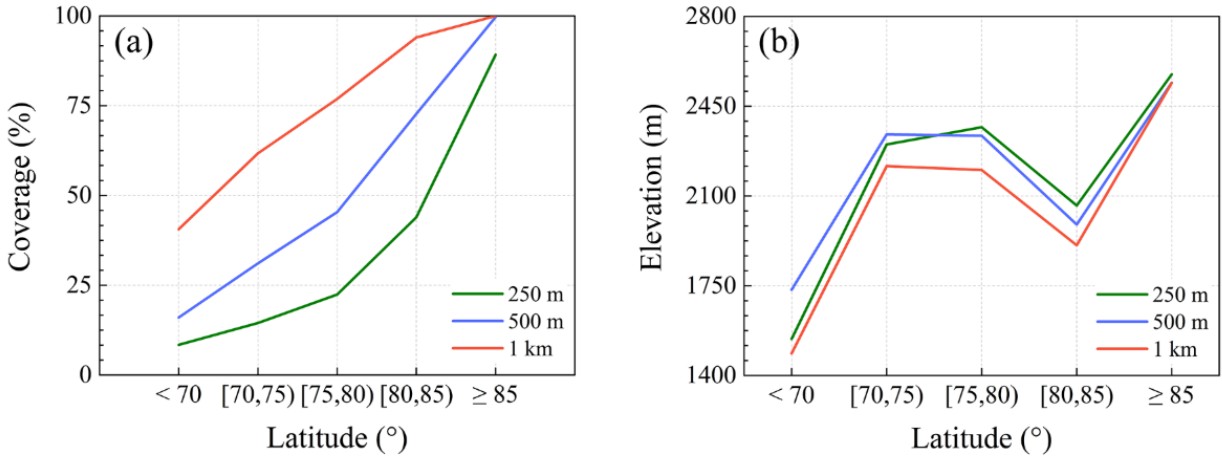

**Figure 3.** (a) Spatial coverages of observed grid cells in the five latitude ranges when three different spatial resolutions, i.e.,
250 m (green), 500 m (blue) and 1 km (red), are applied. (b) Mean elevations in the five latitude ranges when three different
spatial resolutions are applied.

### 2.4.3 DEM evaluation method

Spatiotemporally matched elevation measurements from the OIB airborne mission and ICESat-2 DEM are used to evaluate

the ICESat-2 DEM, and one ICESat-2 DEM grid cell usually has several OIB measurement points. In each grid cell, the

DEM elevation values are subtracted from the median of all OIB elevations within it, and this difference is chosen as the

final bias for each grid cell, which can minimize the effect of abnormal values or outliers.

Four indexes are used to evaluate the DEM performance, including median deviation (MeD), median absolute deviation

(MeAD), standard deviation (SD) and RMSD. The corresponding calculation equations are listed as follows:

$$\text{MeD} = \text{median}(\delta_{i=1,2...,n}) \tag{2}$$

$$\text{MeAD} = \text{median}(|\delta_{i=1,2...,n}|) \tag{3}$$





$$SD = \sqrt{\frac{\sum_{i=1}^{n} (\delta_i - MD)^2}{n-1}}$$

(4)

$$RMSD = \sqrt{\frac{\sum_{i=1}^{n} \delta_i^2}{n-1}}$$

(5)

Where $\delta_i$ is the bias of ICESat-2 DEM and OIB elevation in the grid cell, MD is the mean deviation and $n$ is the number of the matched grid cells.

## 3 Results

### 3.1 General attributes of ICESat-2 DEM

The effective time stamp of the ICESat-2 DEM is May 2019, which is halfway between November 2018 and November 2019. The ICESat-2 DEM provides a complete surface elevation reference for Antarctica, which illustrates higher elevations in the ice sheet interior and lower values in marginal ice shelves (Fig. 4). The local slope shows a pattern similar to the DEM, and undulated slopes are found in areas with rugged terrain, such as the Antarctic Peninsula and Transantarctic Mountains (Fig. 5). Both elevation and slope uncertainties show latitude-dependent patterns, and larger values tend to be found at low latitudes, which may be related to the numbers of ICESat-2 measurement points in individual grid cells (Fig. 2).

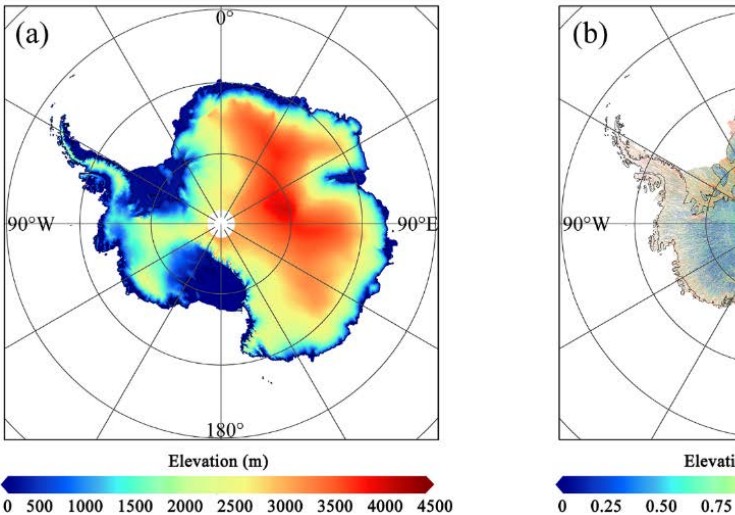

**Figure 4.** (a) A fine-scale DEM of Antarctica at a resolution of 250 m derived from ICESat-2, which covers both the ice sheet and ice shelves with the southern limit of 88°S. (b) Map of the ICESat-2 DEM elevation uncertainty.





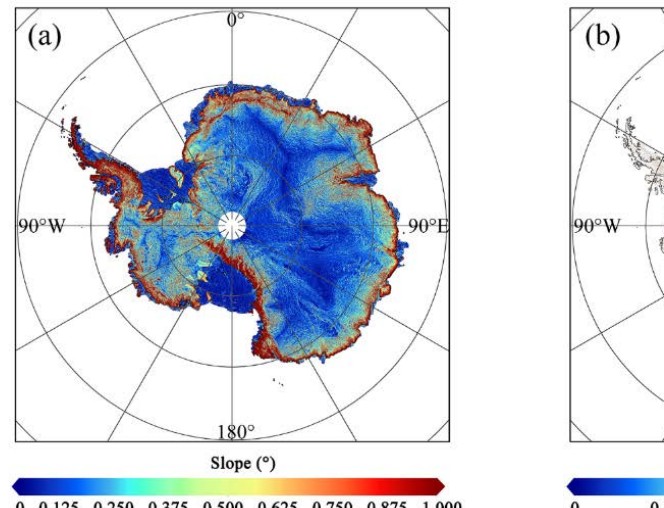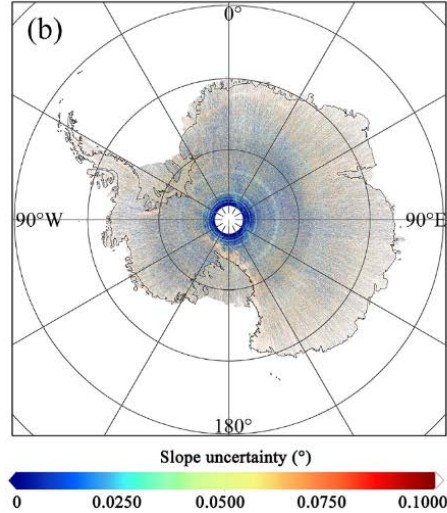

**Figure 5.** (a) Map of the surface slope of Antarctica derived from the ICESat-2 DEM. (b) Map of the ICESat-2 DEM surface
slope uncertainty. The uncertainty is estimated based on the propagation of elevation uncertainty.

   According to the shaded relief map of Antarctica derived from the ICESat-2 DEM (Fig. 6), obvious topographical patterns

and flat terrain can be found in the mountain environments and ice sheet interior, respectively. On the Antarctic Peninsula,

good agreement can be found between the grounding line locations determined from ICESat-2 in Li et al. (2020) and the ice

shelf limit is visually identified from the shaded relief map (Fig. 6b). Other large-scale terrain features, e.g., subglacial lakes

and floating ice shelves, can also be visually detected (Figs. 6c and 6d).



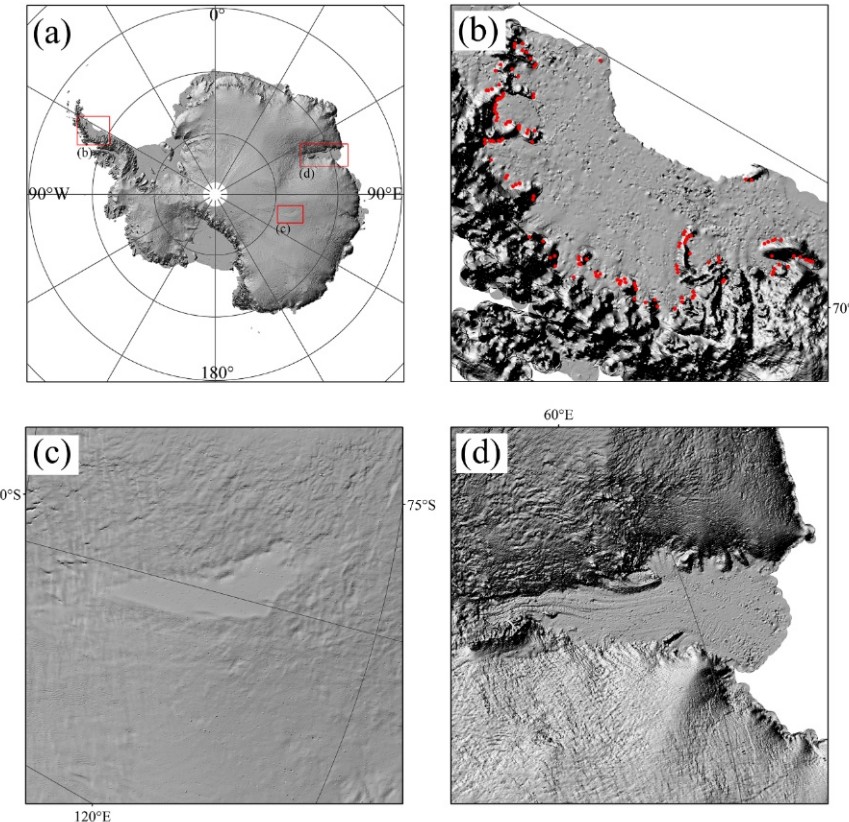

**Figure 6.** (a) Shaded relief map of Antarctica derived from the ICESat-2 DEM. The detailed maps of the Larsen C ice shelf,
Lake Vostok and Amery ice shelf are shown in (b), (c) and (d), respectively, and their locations are also shown in (a) by red
rectangular boxes. The grounding line of the Larsen C ice shelf is marked by red dots in (b), which is mapped from ICESat-2
data given in Li et al. (2020).

Three spatial resolutions are used in the ICESat-2 DEM, and the distributions of four kinds of grid cells (observed at

individual resolutions and interpolated) show obvious latitude-dependent patterns. Regardless of whether at the basin scale

or regional scale, more elevations at higher resolutions tend to be located in high-altitude areas, while elevations at lower or

interpolated resolutions are mostly located in low-altitude regions (Fig. 7).





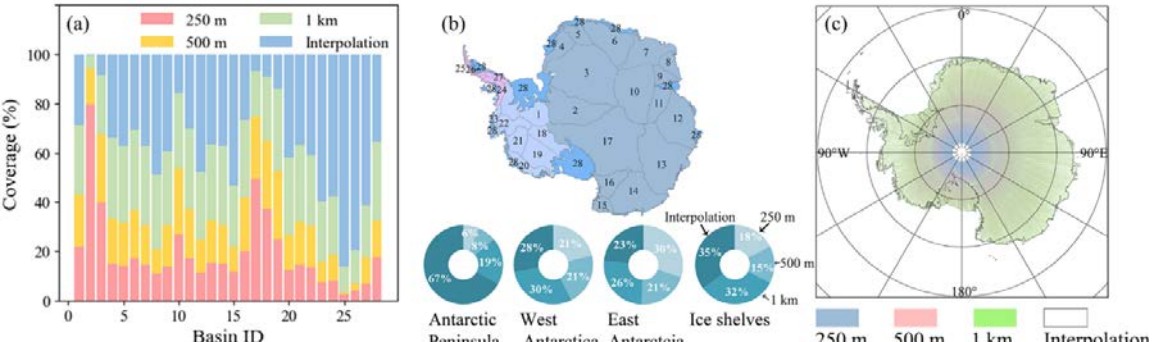

**Figure 7.** (a) Coverages of observed grid cells at 250 m, 500 m and 1 km and interpolated grid cells in 27 drainage basins of ice sheets (Zwally et al., 2012) and ice shelves. The boundaries and basin index (ID) of 27 ice sheet drainage basins (Numbers 1 to 27) and ice shelves (Number 28) are shown in (b). The coverages of observed (at three spatial resolutions) and interpolated grid cells in the Antarctic Peninsula, West Antarctica, East Antarctica and ice shelves are also shown in (b). (c) Map of the selected grid cell resolution for deriving the ICESat-2 DEM in all grid cells at a spatial resolution of 250 m. Elevation values derived from 500 m, 1 km and interpolation (i.e., 1 km) are resampled to a resolution of 250 m.

### 3.2 Evaluation of ICESat-2 DEM by comparing to OIB airborne data

In total, approximately $6 \times 10^9$ OIB measurement points that cover both the steep and flat regions (Fig. 1) are chosen to evaluate the ICESat-2 DEM, and one ICESat-2 DEM grid cell generally contains $17 \pm 9$ OIB measurement points. Generally, the ICESat-2 DEM shows a decimeter-scale bias compared to the OIB surface heights, with a median bias of 0.11 m (Table 3). Ice sheet elevations are more accurate than those estimated for ice shelves.

**Table 3.** Comparisons between the ICESat-2 DEM and spatiotemporally consistent OIB airborne elevation measurements in observed and interpolated areas for individual regions (i.e., the ice sheet and ice shelves).

|  | Region | MeD (m) | MeAD (m) | SD (m) | RMSD (m) | Number of compared grid cells |
|---|---|---|---|---|---|---|
| Observed | Ice sheet | 0.03 | 1.14 | 5.07 | 5.08 | 75687 |
|  | Ice shelves | 0.75 | 1.67 | 6.74 | 6.74 | 7672 |
|  | Total | 0.07 | 1.19 | 5.25 | 5.25 | 83359 |
| Interpolated | Ice sheet | 0.35 | 3.66 | 11.65 | 11.70 | 42922 |
|  | Ice shelves | 0.19 | 4.29 | 11.68 | 11.68 | 6669 |
|  | Total | 0.33 | 3.76 | 11.66 | 11.70 | 49591 |
| Overall | Ice sheet | 0.08 | 1.72 | 8.10 | 8.12 | 118609 |
|  | Ice shelves | 0.63 | 2.52 | 9.37 | 9.37 | 14341 |
|  | Total | 0.11 | 1.80 | 8.25 | 8.27 | 132950 |




We also evaluate the elevation performance for observed and interpolated grid cells (Table 3). Generally, the bias of observed elevations is obviously smaller than that of interpolated elevations in both ice sheets and ice shelves, which indicates that the observed elevations tend to be more accurate than those estimated from interpolation. Larger biases are included in the ICESat-2 DEM if the coverage of interpolated elevations is high, which demonstrates the reasonability of the three resolutions used for DEM generation from ICESat-2. The accuracy of the ICESat-2 DEM has an obvious relationship

with local terrain conditions, and the bias rises when the slope or roughness becomes larger, which is visible for three surface types (Table 4) and three regions with different surface terrains (Fig. 8). The bias in rocks is obviously larger than those for snow/firn and blue ice areas (BIAs), which is mainly due to the local terrain condition, as they are mostly located in the Transantarctic Mountains and the Antarctic Peninsula, while snow/firn and BIAs tend to have flat surface terrain; hence, they have smaller biases. While in the ice sheet interior, the ICESat-2 DEM shows good agreement with the OIB data; in the

Pine Island Glacier region and marginal ice sheet, larger biases occur, which is due to the steep terrain conditions there (Fig. 8).

**Table 4.** Comparison between the ICESat-2 DEM and spatiotemporally consistent OIB airborne elevation measurements with respect to three surface types, i.e., snow/firn, blue ice areas (BIAs) and rocks. The surface type data are obtained from
Hui et al. (2017)

|            | MeD (m) | MeAD (m) | SD (m) | RMSD (m) | Number of compared grid cells |
|------------|---------|----------|--------|----------|-------------------------------|
| Snow/firn  | 0.11    | 1.70     | 7.85   | 7.87     | 127315                        |
| BIA        | -0.14   | 6.73     | 14.35  | 14.35    | 5494                          |
| Rock       | -4.74   | 16.17    | 23.43  | 23.43    | 141                           |
| Total      | 0.11    | 1.80     | 8.25   | 8.27     | 132950                        |


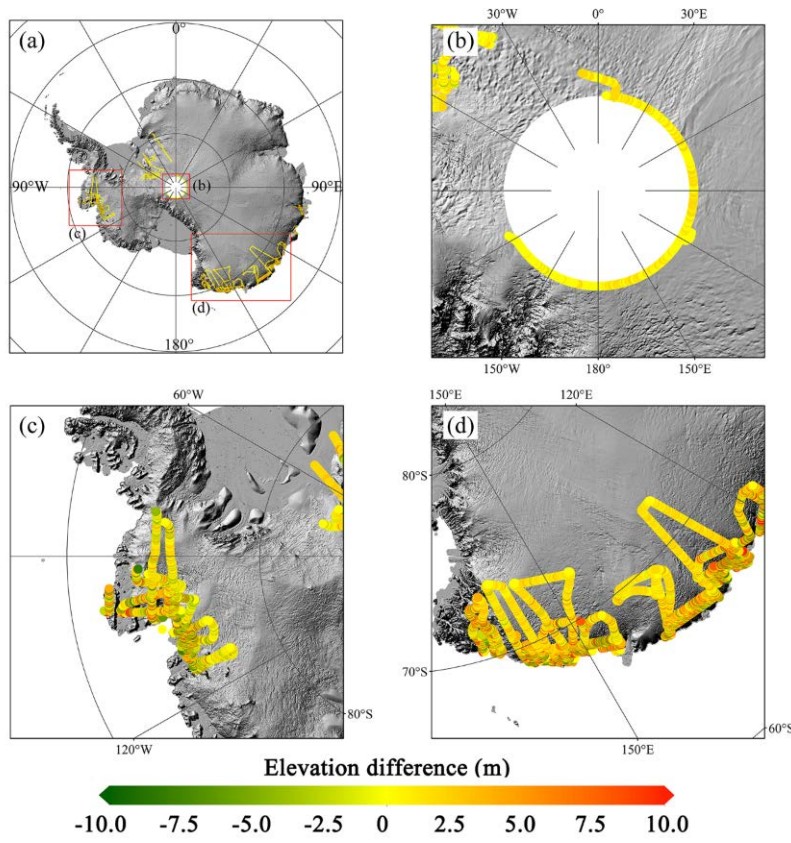

**Figure 8.** (a) Map of the difference between the ICESat-2 DEM and OIB airborne elevation measurements. Detailed maps of the ice sheet interior, Pine Island Glacier region and marginal ice sheet in East Antarctica are shown in (b), (c) and (d), and their locations are also shown in (a) by red rectangular boxes. The background is the shaded relief map of Antarctica derived from the ICESat-2 DEM.

Although OIB airborne data provide an independent evaluation of the generated DEM, they still cannot present a comprehensive comparison. Most of the OIB airborne data were obtained in ice sheet margins or mountain environments, with high slopes and low elevations. Approximately 74% of OIB elevations are less than 1500 m, and 74% of the observed surface slopes from the OIB mission are less than 1° (Fig. 1), while the corresponding percentages from the ICESat-2 DEM are 37% and 89%, respectively. The applied OIB airborne data cannot completely represent the slope/elevation distributions of the Antarctic DEM; hence, the real accuracy of the ICESat-2 DEM is biased and may be higher.

## 4 Comparisons with previous published Antarctic DEMs

When compared to the altimeter-derived DEMs, the elevation difference rises when the surface slope becomes larger, especially in mountainous environments (e.g., Transantarctic Mountains and Antarctic Peninsula, Fig. 9). This may be due to



their differences in spatial resolution and measurement accuracy; this effect is considerably reduced when the local terrain is flatter (e.g., ice sheet interior).

Compared to the REMA DEM and TanDEM PolarDEM, smaller elevation differences can be found in both the flat ice
sheet interior and steep mountains/marginal ice sheets. Usually, DEMs derived from stereo-photogrammetry have a better performance for high-slope regions (Slater et al., 2018); hence, similar elevations indicate the reliability of ICESat-2 DEMs in mountain environments. In particular, the ICESat-2 DEM shows a generally higher surface height than the TanDEM PolarDEM, which is assumed to be caused by the penetration depth of the X-band (TerraSAR-X and TanDEM-X) into snowpack (Fischer et al., 2020; Dehecq et al., 2016).


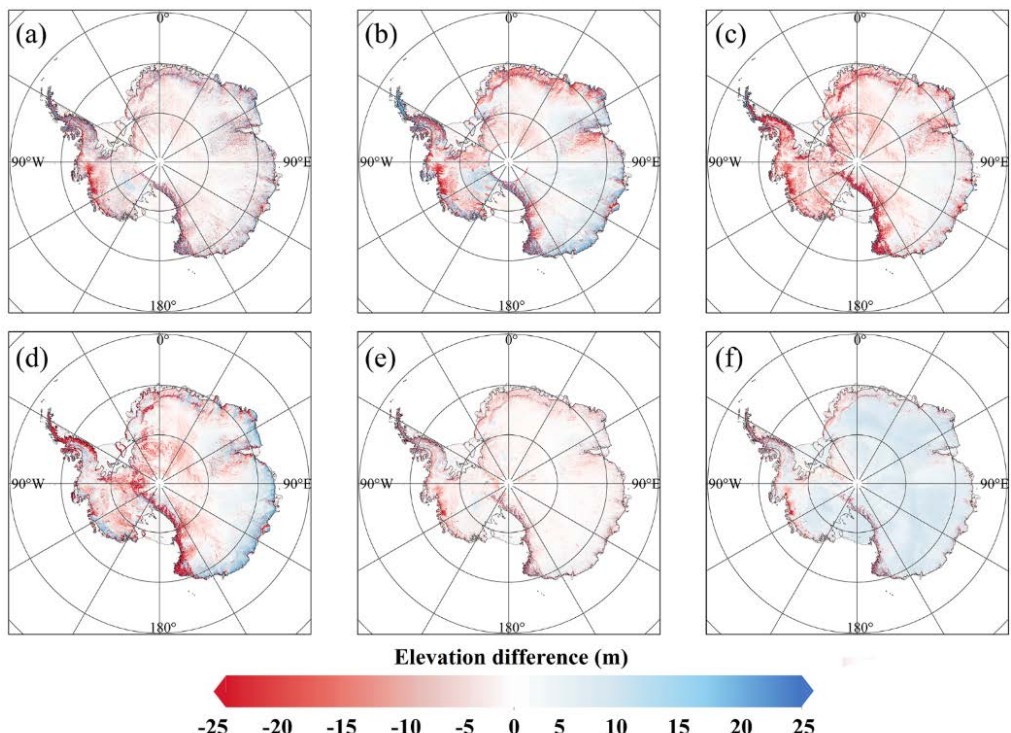

**Figure 9.** Elevation differences between the ICESat-2 DEM and six previously published DEMs, i.e., ICESat DEM (a), ICESat/ERS-1 DEM (b), Helm CryoSat-2 DEM (c), Slater CryoSat-2 DEM (d), REMA DEM (e) and TanDEM PolarDEM (f).


To indicate a quantitative comparison between the ICESat-2 DEM and other DEMs, OIB airborne data are used to evaluate individual DEMs, and the same evaluation method applied for the ICESat-2 DEM is used (as described in Section 2.2). The ICESat DEM and ICESat/ERS-1 DEM are not included here, as no spatiotemporally matched OIB data can be





found. The evaluation result shows that the ICESat-2 DEM has a better performance than altimeter-derived DEMs and is

comparable to the DEMs derived from stereo-photogrammetry and interferometry (Table 5).

**Table 5.** Comparisons between the ICESat-2 DEM, Helm CryoSat-2 DEM, Slater CryoSat-2 DEM, REMA DEM, TanDEM PolarDEM and corresponding spatiotemporally consistent OIB airborne elevation measurements.

|  | MeD (m) | MeAD (m) | SD (m) | RMSD (m) | Number of compared grid cells |
|---|---|---|---|---|---|
| ICESat-2 DEM | 0.11 | 1.80 | 8.25 | 8.27 | 132950 |
| Helm CryoSat-2 DEM | 1.30 | 4.72 | 12.68 | 13.38 | 23026 |
| Slater CryoSat-2 DEM | -0.26 | 2.51 | 10.42 | 10.42 | 39100 |
| REMA DEM | 0.16 | 0.95 | 3.86 | 3.89 | 319236 |
| TanDEM PolarDEM | -1.76 | 2.66 | 6.29 | 6.47 | 890 |

The median differences in surface slope and roughness for these five DEMs illustrate that all their elevation biases become more uncertain with increasing slope and roughness (Fig. 10). The ICESat-2 DEM outperforms other altimeter-derived DEMs for all surface conditions. The REMA DEM always has more stable performances than the ICESat-2 DEM, as stereo-photogrammetry can generate more consistent elevation estimations at the regional scale than altimetry. A similar situation occurs for the TanDEM PolarDEM under most surface conditions (slopes >1° or roughnesses >5 cm). Nevertheless, the

ICESat-2 DEM is comparable to both the REMA DEM and TanDEM PolarDEM when slopes are less than 1°, which occupies 89% of Antarctica north of 88°S. In addition, compared to the REMA DEM, the ICESat-2 DEM can provide an elevation reference with a definite time stamp, which is essential for further ice dynamics and mass change estimation.

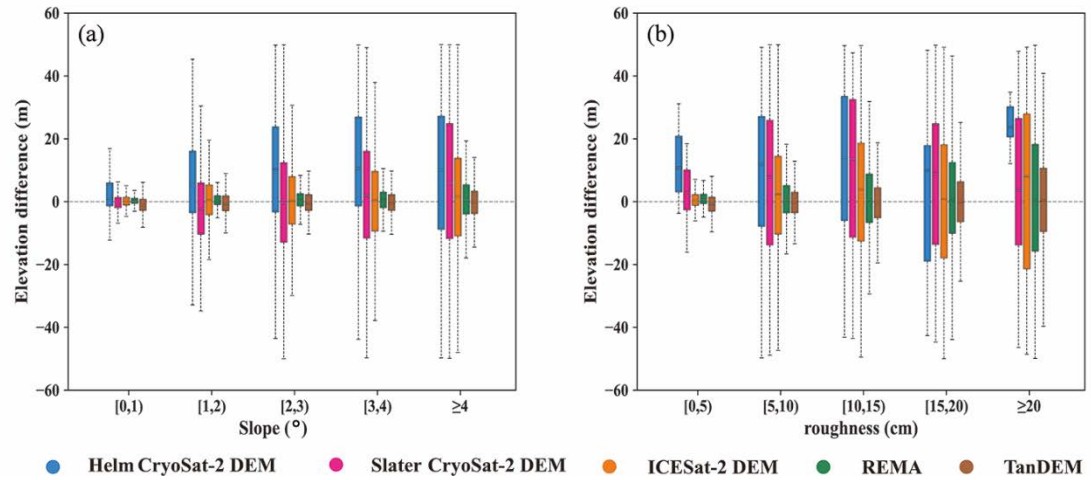

**Figure 10.** Median differences between five DEMs and corresponding spatiotemporally consistent OIB airborne elevation
measurements for surface slope and roughness. The upper and lower lines in each box indicate the 25th and 75th percentiles, the whiskers indicate the 5th and 95th percentiles, and the central horizontal line indicates the median difference.



Here, OIB airborne data from 2009 to 2019 in rocky regions are used to construct a comprehensive evaluation. The basic assumption is that the elevation changes in the rocks between 2009 and 2019 are negligible, and the ICESat DEM and ICESat/ERS-1 DEM are also included. As surfaces in rocky regions are usually steep, stereo-imagery-derived DEMs show better performance, and the ICESat-2 DEM outperforms all altimeter-derived DEMs (Table 6).

**Table 6.** Comparisons between seven DEMs (including the ICESat-2 DEM and six previously published DEMs) and corresponding spatiotemporally consistent OIB airborne elevation measurements in rocky regions.

|  | MeD (m) | MeAD (m) | SD (m) | RMSD (m) | Number of compared grid cells |
|---|---|---|---|---|---|
| ICESat-2 DEM | -1.02 | 10.06 | 21.07 | 21.14 | 2277 |
| ICESat DEM | 1.93 | 13.75 | 23.52 | 23.72 | 1072 |
| ICESat/ERS-1 DEM | 0.44 | 16.63 | 24.03 | 24.01 | 582 |
| Helm CryoSat-2 DEM | 2.59 | 16.93 | 24.44 | 25.00 | 428 |
| Slater CryoSat-2 DEM | -0.81 | 16.36 | 24.68 | 24.66 | 457 |
| REMA DEM | 0.57 | 3.74 | 13.86 | 13.87 | 4086 |
| TanDEM PolarDEM | 1.25 | 4.01 | 11.86 | 11.86 | 554 |

## 5 Conclusions

A definite time-stamped (May 2019) DEM for Antarctica with a modal resolution of 250 m is presented based on the surface height measurements from ICESat-2 by using a model fitting method. This DEM has an elevation measurement that accounts for 74% of Antarctica, and the remaining 26% is estimated based on the ordinary kriging method. The accuracy of the ICESat-2 DEM is evaluated by comparing it to the independent airborne data from the OIB mission after spatiotemporal matching. Overall, the ICESat-2 DEM shows a median bias of 0.11 m and an RMSD of 8.27 m, and these accuracies are compromises for DEM values from surface fits and interpolation. A median bias of 0.07 m and an RMSD of 5.25 m are found for areas where elevations are derived from ICESat-2 measurements, and they increase to 0.33 m and 11.70 m for interpolated elevations. The accuracy decreases when the surface slope or roughness increases; thus, larger biases occur for steep rocks, and flat snow/firn and blue ice areas have smaller elevation differences.

Compared to DEMs derived from satellite altimeters (i.e., the ICESat DEM, ICESat/ERS-1 DEM, Helm CryoSat-2 DEM, and Slater CryoSat-2 DEM), larger differences are found in regions with high slopes, which is due to their resolution difference, while smaller elevation differences compared to the REMA DEM and TanDEM PolarDEM support the reliability of the ICESat-2 DEM, as DEMs derived from stereo-photogrammetry and interferometry usually have better performances in steep areas. The ICESat-2 DEM shows better performance than altimeter-derived DEMs and is comparable to the fine-



scale REMA DEM and TanDEM PolarDEM, which demonstrates the reliability of the ICESat-2 DEM. More importantly, the ICESat-2 DEM has a specific time stamp, which is more valuable for further scientific applications, e.g., land ice height and mass balance estimations.

**Data availability**

The ICESat-2 DEM (including map of uncertainty) generated in this paper is available at https://data.tpdc.ac.cn/en/disallow/9427069c-117e-4ff8-96e0-4b18eb7782cb/ temporarily, for now the data archiving is underway and this link will be replaced by an official DOI link after review; ICESat-2 data are available at https://nsidc.org/data/icesat-2; OIB airborne data are derived from https://nsidc.org/data/ILATM2/versions/2.

**Author contribution**

Xiaoyi Shen and Yubin Fan developed the related algorithm, generated and evaluated the ICESat-2 DEM; Lhakpa Drolma constructed the comparison to previously published DEM products; Chang-Qing Ke supervised this work.

**Competing interests**

The authors declare that they have no conflict of interest

**Acknowledgments**

This work is supported by the Programs for National Natural Science Foundation of China [grant numbers 41976212, 41830105].

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
