# Peer review of "A fine-scale digital elevation model of Antarctica derived from ICESat-2"

_The Cryosphere, 2021_

## Referee Comment (RC1)

Review of Shen et al, TC 2021

This paper presents a new DEM of Antarctica derived from a year's worth of ICESat-2 data. It is a worthwhile endeavour and, with adequate care and attention to detail, an ICESat-2 DEM could provide a useful additional source of topographic information for Antarctica. There are, however, some misgivings with the approach taken here and lack of clarity about some of the methods.

General comments

In general the paper is well written but there are a significant number of instances of statements that are not accurate and/or misleading possibly because of the terminology used or possibly because of flaws in understanding or misconceptions. Specific details are given below but they start with the very first sentence of the abstract and continue from there and occur frequently.

If I understand the approach correctly, the authors produced a DEM at 250 km posting by resampling DEMs at 500 m and 1 km posting. First, the authors confuse resolution and posting (they are not the first to do this). It is misleading or incorrect to claim that a DEM with 26% observed coverage at 250 m posting has a resolution of 250 m as 74% of the grid points are interpolated. It has a 250 m posting and a resolution that is latitudinally dependent. More serious and worrying appears to the fact that there is a bias between elevations interpolated at the three resolutions which amounts to ~100 m at different latitudes (Fig 3b). According to this Fig. the mean elev at, say, 80 degs differs by ~150 m from 1 km to 250 m. if the interpolation has been done correctly what is the explanation for this huge difference and how is it possible to combine elevations at these three resolutions when they are so different? I must have missed something but this seems like a fundamental issue? It suggests the distribution of elevations is not Gaussian around the mean, which is entirely possible but the interpolation should account for this unless there is a flaw in the method used.

A second major reservation, which in addition contradicts the brief discussion of the results, is that the largest mean elevation difference with OIB data, standard dev and RMSD is found over the ice shelves. These are the flattest places in Antarctica and so it is worrying and extremely unexpected that the "worst" bias and random error is over the ice shelves. This makes no sense. Without some sound explanation for why this would be, it brings into question the methodology/approach used here. Combined with the strange bias between different resolutions this suggest some issues with the methodology.

Third. The DEM is presented and compared to six existing DEMs but there is no real attempt to discuss what value or use it might have compared to these, how and where it might be better/worse than one or other or why someone might want to use this DEM over, for example, the REMA DEM which has an RMSD that is half of the ICESat-2 one. It could be better for some applications but no attempt is made to consider what these might be. Was this DEM created because it could be or because it should be? The former has limited scientific value, the latter needs to have demonstrated scientific value.

Specific Comments

L9-10. "ice topography monitoring and ice mass change estimation". First, these two things are almost identical, the difference being a density, so to assert that they are somehow different is misleading. To get mass balance you have to measure an elevation change here. Second, the vast majority of altimetry-derived mass balance and "topography monitoring" approaches published do not use a DEM (partly because of biases in absolute elevation) but use elevation difference either at cross overs or along track.

L10 thirty decades = 300 years.

L24 "essential addition". This is not demonstrated in the m/s. While the new DEM is an interesting addition to the 6 discussed, it is not the most accurate so to claim it is essential is unjustified.

L27-28. See first comment above. In addition, the two references cited do not use altimetry DEMs for ice motion tracking or mass balance.

L30 "monitoring the topography". What do the authors mean? Do they mean measuring elevation change? See first comment.

L51-52. KU-band penetration into snowpack is unknown. This statement is misleading at best. There is plenty of literature on Ku band penetration into snow. Further, radar penetration can be corrected for either empirically or theoretically using a waveform fitting approach (see e.g. [*Davis*, 1996; 1997]).

L84 Icessn?

L87-88. This is not reducing "seasonal elevation changes" because any change that occurs in less than a year must be sub-annual, not multi-annual. Second, except for the pensinsula seasonal changes do not really exist over Antarctica, they are sub-annual which is not the same thing.

Table 1, p5. It states the the ICESat/ERS-1 DEM has an uncelar timestamp but in the paper there is a specific section the DEM time stamp, where it explains that the ICESat data are corrected for any significant dh/dt between 1995 and date of acquisition [*Bamber et al.*, 2009] so the time stamp is extremely clear and spelled out. It may not be perfectly corrected to 1995 but the Table is incorrect for this DEM and, consequently, makes me concerned about the veracity of claims made elsewhere in Table 1. Without carefully studying the methods used to generate each DEM the authors will not be able to support the claims made in Table 1.

L120 replace "seasonal" with sub annual here and elsewhere.

L130 neighboring. The preferred spelling convetion for TC is UK English not US. Please follow that convention.

L153. Strictly speaking the kriging variance is not, and cannot, be used to determine the interpolation error. It is related to the interpolation error but not equal to it.

L171. Nominal resolution. See above. It would be more appropriate to refer to 250 m posting with a resolution of 1 km for latitudes x-y and 250 m from a-b…

L248. Needs rewording.

L250 Here the authors claim the bias increases with slope or roughness but that is not borne out by Table 3. See General Comments.

L280-281. This is a sweeping generalisation that is incorrect. The accuracy of photogrammetric derived DEMs is a function of the resolution of the sensor and the accuracy of GCPs.

L321. "modal resolution" -> posting.

References

Bamber, J. L., Gomez Dans, J. L., and Griggs, J. A. (2009), A new 1 km digital elevation model of the Antarctic derived from combined satellite radar and laser data. Part I: Data and methods, *The Cryosphere 3*(2), 101-111.

Davis, C. H. (1996), Temporal change in the extinction coefficient of snow on the Greenland ice sheet from an analysis of seasat and geosat altimeter data, *IEEE Trans. Geosci. Remote Sensing*, *34*(5), 1066-1073.

Davis, C. H. (1997), A robust threshold retracking algorithm for measuring ice-sheet surface elevation change from satellite radar altimeters, *IEEE Trans. Geosci. Remote Sensing*, *35*(4), 974-979.

---

## Author Comment (AC1)

We thank the reviewer for the helpful feedback, these suggestions have significantly improved the text and figures, we are appreciative of the help and time.

We have addressed all the comments here, point by point responses to the comments are listed in BLUE.

Here we summarize two major revisions in the revised manuscript:

First, in the revised manuscript we select a 500m posting for the derived ICESat-2 DEM. That is, elevations are firstly estimated at the resolution of 500 m and 1 km, respectively. Then the observation gaps in the 500 m DEM are filled by the resampled 1 km DEM (resampled to the 500 m), and the remaining gaps are filled by using ordinary kriging interpolation. In the original manuscript, we firstly generated four DEMs, i.e., 250m DEM, 500m DEM, 1km DEM and interpolated DEM. We produced a DEM at 250 m posting by resampling DEMs at 500 m and 1 km posting (including the interpolated DEM). However, 250m DEM only has a coverage of 26%, most of the derived DEM were based on resampled coarser DEMs. Hence, as suggested by the reviewers, starting from 500m, refill with 1km and do the kriging would make more sense. In this case the model fitting tends to be more robust as more points are used per grid cell. The updated Figure 4 is listed below (the figure indexes in this text are the same to these in original manuscript):

[Figure]

**Figure 4.** (a) A fine-scale DEM of Antarctica at a posting of 500 m derived from ICESat-2, which covers both the ice sheet and ice shelves with the southern limit of 88°S. (b) Map of the ICESat-2 DEM elevation uncertainty.

Second, in order to provide a robust and reasonable comparison between ICESat-2 DEM and other DEMs, a common OIB data in areas of low elevation change from 2009 to 2019 are used in the revised manuscript. As the same evaluation data are used, all the DEMs can be compared validly. These OIB data are located in the ice sheet interior, as shown in the figure below (Figure 1b). The CryoSay-2 Low Rate Mode (which was designed for flat ice sheet interior measurements) mask is used to extract the regions of low elevation change. CryoSat Geographical Mode Mask (v 4.0, updated in 19 August - 26 August 2019) at https://earth.esa.int/eogateway/news/cryosat-geographical-mode-mask-4-0-released is used here. The averaged elevation change rate in the used

OIB data locations is about -0.0074±0.0821 m/yr from 2003 to 2019, according to elevation change rate data from Smith et al. (2020). Hence, we assume that in these areas the effect of the elevation change on the DEM evaluation can be ignored. It should be noted that, when evaluating ICESat-2 DEM individually, OIB data in areas of low elevation change from 2009 to 2017 and OIB data in all Antarctica from 2018 to 2019 (Figure 1a) are used for a comprehensive evaluation. In addition, published GPS transects (Schröder et al., 2017) are also used here for the DEM comparison (Figure 1c).

[Figure]

Elevation (m)

0   500   1000   1500   2000   2500   3000   3500   4000

**Figure 1.** (a) Map of the OIB airborne data in October and November 2018 and October and November 2019 in Antarctica. (b) Map of the OIB airborne data from 2009 to 2019 in Antarctic ice sheet interior. (c) Map of the GPS transects from 2001 to 2015 in Antarctica. The dashed lines in (b) and (c) show the boundary of region where we assume to have low elevation change, it is the mode mask boundary of CryoSat-2 Low Rate Mode data in Antarctica, this data mode was designed for flat ice sheet interior measurements.

In addition, in the revised manuscript we resample all the DEMs to the OIB data locations and calculate the difference and statistics to perform the evaluation. The original approach (calculating a median OIB elevation for each grid cell) will certainly influence the evaluation results as the DEMs have different pixel spacing. Also as pointed out by the reviewers, OIB is the reference elevation and cannot be replaced by the median values. The new approach is also applied for DEM evaluation using GPS data. The related statistics have changed, but the same conclusions are found comparing to original statistics. The updated Tables 3, 5 and 6 are listed in below:

**Table 3.** Comparisons between the ICESat-2 DEM and OIB airborne elevation measurements (including data in areas of low elevation change from 2009 to 2017 and data in all Antarctica from 2018 to 2019) in observed and interpolated areas for individual regions (i.e., the ice sheet and ice shelves).

|  | Region | MeD (m) | MeAD (m) | SD (m) | RMSD (m) | Number of used OIB measurement points |
|---|---|---|---|---|---|---|
| Observed | Ice sheet | 0.08 | 1.18 | 12.75 | 12.75 | 3589087 |
|  | Ice shelves | 0.77 | 2.60 | 15.26 | 15.27 | 191754 |
|  | Total | 0.09 | 1.23 | 12.89 | 12.89 | 3780841 |
| Interpolated | Ice sheet | -0.40 | 2.50 | 20.68 | 20.73 | 1237416 |
|  | Ice shelves | 0.36 | 3.23 | 24.61 | 24.65 | 185613 |

| | | | | | |
|---|---|---|---|---|---|
| | Total | -0.33 | 2.58 | 21.25 | 21.28 | 1423029 |
| Overall | Ice sheet | 0.01 | 1.41 | 15.20 | 15.20 | 4826503 |
| | Ice shelves | 0.59 | 2.88 | 20.40 | 20.43 | 377367 |
| | Total | 0.03 | 1.49 | 15.64 | 15.64 | 5203870 |

**Table 5.** Comparisons between the ICESat-2 DEM, ICESat DEM, ICESat/ERS-1 DEM, Helm CryoSat-2 DEM, Slater CryoSat-2 DEM, REMA DEM, TanDEM PolarDEM and OIB airborne elevation measurements in areas of low elevation change from 2009 to 2019.

| | MeD (m) | MeAD (m) | SD (m) | RMSD (m) | Number of used OIB measurement points |
|---|---|---|---|---|---|
| ICESat-2 DEM | 0.10 | 0.98 | 5.36 | 5.38 | |
| ICESat DEM | -2.61 | 6.35 | 19.90 | 20.43 | |
| ICESat/ERS-1 DEM | -0.15 | 1.84 | 11.53 | 11.54 | |
| Helm CryoSat-2 DEM | 0.65 | 2.68 | 24.97 | 25.02 | 1965309 |
| Slater CryoSat-2 DEM | 1.22 | 2.87 | 23.85 | 24.14 | |
| REMA DEM | -0.16 | 0.53 | 1.75 | 1.76 | |
| TanDEM PolarDEM | -2.84 | 2.94 | 2.76 | 3.90 | |

**Table 6.** Comparisons between the ICESat-2 DEM, ICESat DEM, ICESat/ERS-1 DEM, Helm CryoSat-2 DEM, Slater CryoSat-2 DEM, REMA DEM, TanDEM PolarDEM and GPS elevation data in areas of low elevation change from 2001 to 2015.

| | MeD (m) | MeAD (m) | SD (m) | RMSD (m) | Number of used OIB measurement points |
|---|---|---|---|---|---|
| ICESat-2 DEM | -0.03 | 0.41 | 1.17 | 1.17 | |
| ICESat DEM | -1.91 | 2.89 | 5.21 | 5.97 | |
| ICESat/ERS-1 DEM | -0.74 | 0.84 | 1.39 | 1.61 | |
| Helm CryoSat-2 DEM | 0.07 | 0.67 | 1.67 | 1.71 | 488963 |
| Slater CryoSat-2 DEM | 0.00 | 0.46 | 1.65 | 1.66 | |
| REMA DEM | 0.03 | 0.26 | 0.57 | 0.57 | |
| TanDEM PolarDEM | -4.62 | 4.62 | 1.33 | 4.72 | |

References:

Smith B, Fricker H A, Gardner A S, et al. Pervasive ice sheet mass loss reflects competing ocean and atmosphere processes. Science, 2020, 368(6496): 1239-1242.

Schröder L, Richter A, Fedorov D V, et al. Validation of satellite altimetry by kinematic GNSS in central East Antarctica. The Cryosphere, 2017, 11(3): 1111-1130.

Review of Shen et al, TC 2021

This paper presents a new DEM of Antarctica derived from a year's worth of ICESat-2 data. It is a worthwhile endeavour and, with adequate care and attention to detail, an ICESat-2 DEM could provide a useful additional source of topographic information for Antarctica. There are, however, some misgivings with the approach taken here and lack of clarity about some of the methods.

General comments

In general the paper is well written but there are a significant number of instances of statements that are not accurate and/or misleading possibly because of the terminology used or possibly because of flaws in understanding or misconceptions. Specific details are given below but they start with the very first sentence of the abstract and continue from there and occur frequently.

Thank you very much for your feedback and advice, these suggestions have significantly improved the text and figures. We have revised the manuscript according to all your comments, details are listed in below.

If I understand the approach correctly, the authors produced a DEM at 250 km posting by resampling DEMs at 500 m and 1 km posting. First, the authors confuse resolution and posting (they are not the first to do this). It is misleading or incorrect to claim that a DEM with 26% observed coverage at 250 m posting has a resolution of 250 m as 74% of the grid points are interpolated. It has a 250 m posting and a resolution that is latitudinally dependent. More serious and worrying appears to the fact that there is a bias between elevations interpolated at the three resolutions which amounts to ~100 m at different latitudes (Fig 3b). According to this Figure the mean elevation, say, 80 degs differs by ~150 m from 1 km to 250 m. if the interpolation has been done correctly what is the explanation for this huge difference and how is it possible to combine elevations at these three resolutions when they are so different? I must have missed something but this seems like a fundamental issue? It suggests the distribution of elevations is not Gaussian around the mean, which is entirely possible but the interpolation should account for this unless there is a flaw in the method used.

1. Thank you very much for this essential correction. We have corrected this misleading terminology in the related part of the manuscript.

2. In the Fig. 3, elevations at different latitudes are presented by the region-averaged values. As shown in the Fig. 2 (in below), the coverages of DEMs at three resolutions are different, the observed elevation grid cells at the same region are not completely overlapped, DEM at coarser resolution has more observed grid cells than DEM at fine resolution at the same region, hence the region-averaged elevation values have large difference.

[Figure]

**Figure 2.** Map of the observed grid cells of DEMs at the spatial resolution of 250 m (a), 500 m (b)

and 1 km (c). The observed grid cells are colored in blue, the overall coverage of each DEM in Antarctica is also presented beside.

In the revised manuscript, we compare the elevations at two resolutions (i.e., 500 m and 1 km, as we mentioned in the very beginning only two resolutions are used in the revised manuscript) in the overlapped regions in Antarctica. The map and histogram of elevation difference between 1km DEM and 500m DEM are shown in Figs. 3b and 3c. An averaged elevation difference of 0.04 ± 2.93 m is found, which is quite small comparing to the estimated elevations. This elevation difference is acceptable and it is valid to combine elevations at these two resolutions.

[Figure]

**Figure 3.** (a) Spatial coverages of observed grid cells in the five latitude ranges when two spatial resolutions, i.e., 500 m (blue) and 1 km (red), are applied. (b) Map of the elevation difference of DEMs at the resolutions of 1 km and 500 m. (c) Histograms of the elevation difference of DEMs at the resolutions of 1 km and 500 m, the average and standard deviation values are also presented.

A second major reservation, which in addition contradicts the brief discussion of the results, is that the largest mean elevation difference with OIB data, standard dev and RMSD is found over the ice shelves. These are the flattest places in Antarctica and so it is worrying and extremely unexpected that the "worst" bias and random error is over the ice shelves. This makes no sense. Without some sound explanation for why this would be, it brings into question the methodology/approach used here. Combined with the strange bias between different resolutions this suggest some issues with the methodology.

As we mentioned in the very beginning, we generate a new DEM at 500 m posting, perform a new evaluation by resampling the DEM into OIB location and calculating the statistics. According to the updated evaluation result, we can still find a slight better performance of DEM in ice sheet than ice shelves. In order to find the explanation, we present the histograms of surface slope and roughness values (derived from OIB data) for ice sheet and ice shelves in below:

[Figure]

**Figure.** Histograms of the OIB-derived surface slope and roughness values for ice sheet and ice shelves.

As we can found in this figure, observed ice shelves have overall smaller surface roughness than ice sheet, but have a larger percentage of high-slope areas than ice sheet. For example, approximately 70% of the OIB measurement points which covered ice sheet have slope values of < 0.01°. In comparison, approximately 50% of the OIB measurement points which located in ice shelves have slope values of < 0.01°. Hence, observed ice shelves have a higher percentage of high-slope areas, which may cause larger elevation biases. To test this argument, standardized regression coefficients between surface slope/roughness and the elevation difference (i.e., mean absolute difference between ICESat-2 DEM and OIB elevations) are calculated here by using a multivariate linear regression model (this model is fitted by using an iterative least-squares fit). All OIB data in 2018 and 2019 are used. Standardized values of surface slope, roughness and elevation difference are used for a valid comparison. The regression coefficients for surface slope and roughness are 0.18 and -0.01. Larger regression coefficient indicates that the surface slope has greater effect on elevation difference than roughness. Hence, although ice shelves observed by OIB data have smaller surface roughness than ice sheet, a higher percentage of high-slope areas makes ice shelves have a slight worse DEM performance.

In addition, similar phenomenon can still be found in Table 2 in Slater et al. (2018), their observed DEM also has a slight better performance in ice sheet than ice shelves.

**Table 2.** Statistics of the comparison between observed and interpolated DEM grid cells and airborne elevation measurements for individual Antarctic regions and mode mask areas. In total, only 5 and 2 % of DEM elevation values are obtained through interpolation for the ice sheet and ice shelves, respectively.

| Region | Observed | | | Interpolated | | |
|---|---|---|---|---|---|---|
| | Number of compared grid cells | Median difference (m) | rms difference (m) | Number of compared grid cells | Median difference (m) | rms difference (m) |
| Ice sheet | 230 165 | −0.27 | 13.36 | 32 933 | 25.37 | 138.62 |
| Ice shelves | 40 081 | −0.42 | 14.31 | 4772 | 1.20 | 30.96 |
| Antarctic Peninsula | 6820 | −1.12 | 22.40 | 7473 | 82.21 | 191.07 |
| West Antarctica | 60 452 | −0.86 | 11.43 | 8783 | 11.78 | 96.15 |
| East Antarctica | 162 893 | −0.17 | 13.60 | 14 679 | 19.62 | 117.77 |
| LRM | 73 867 | 0.26 | 7.15 | 1683 | 6.51 | 41.70 |
| SARIn (ice sheet only) | 156 298 | −0.82 | 15.45 | 31 250 | 28.65 | 141.97 |
| Total | 270 246 | −0.30 | 13.50 | 37 655 | 19.84 | 131.13 |

Reference:

Slater T, Shepherd A, McMillan M, et al. A new digital elevation model of Antarctica derived from CryoSat-2 altimetry. The Cryosphere, 2018, 12(4): 1551-1562.

Third. The DEM is presented and compared to six existing DEMs but there is no real attempt to discuss what value or use it might have compared to these, how and where it might be better/worse than one or other or why someone might want to use this DEM over, for example, the REMA DEM which has an RMSD that is half of the ICESat-2 one. It could be better for some applications but no attempt is made to consider what these might be. Was this DEM created because it could be or because it should be? The former has limited scientific value, the latter needs to have demonstrated scientific value.

Here we add the related discussion in Section 5 in the revised manuscript:

Generally, the accuracy of ICESat-2 DEM is comparable to these of REMA DEM and TanDEM PolarDEM, which have higher resolutions and accuracies. However, these commercial images (used in REMA DEM and TanDEM PolarDEM) are expensive, significant economic investments are required to derive a large-scale DEM (e.g., for Antarctica) as a large amount of data are needed. More importantly, in this case multi-temporal images are usually used, the derived DEMs hence have various time stamps in different regions, which may cause some uncertainties in their further applications, e.g., numerical ice sheet modelling. ICESat-2 DEM has the same time stamp for all regions. Although the derived ICESat-2 DEM is less accurate than REMA DEM and TanDEM PolarDEM, considering the measurement accuracy of altimetry, these differences are still acceptable. Elevation change rate can be obtained when deriving the ICESat-2 DEM, which can provide an additional reference for ice topography and mass balance estimation. Comparing to altimeter-derived DEMs, ICESat-2 DEM has better (or comparable) performance in accuracy, resolution and coverage.

In addition, ICESat-2 data are easily accessible by the public comparing to high-resolution commercial images. In previous studies, several years of altimeter data are needed to derive the DEM in Antarctica. Due to the high-density measurements of ICESat-2, 13 months of ICESat-2 data can be used to generate a DEM for Antarctica, and the elevation accuracy is superior than other altimeter-derived DEMs. This means that the ICESat-2 DEM can be updated annually, this is still difficult for those DEMs derived from stereo-photogrammetry and interferometry. This study demonstrates the feasibility and reliability of using one-year ICESat-2 data to derive the Antarctic DEM, provides a reference for the processing scheme of DEM (e.g., in higher resolution, regularly updated) based on ICESat-2 in future.

Specific Comments

L9-10. "ice topography monitoring and ice mass change estimation". First, these two things are almost identical, the difference being a density, so to assert that they are somehow different is misleading. To get mass balance you have to measure an elevation change here. Second, the vast

majority of altimetry-derived mass balance and "topography monitoring" approaches published do not use a DEM (partly because of biases in absolute elevation) but use elevation difference either at cross overs or along track.

Agree and accept, we have revised this sentence to avoid the existing misleading information: 'Antarctic digital elevation models (DEMs) are essential for human fieldwork, ice motion tracking and the numerical modelling of ice sheet'.

The original sentence is also listed below:

L10 thirty decades = 300 years.

It should be 30 years, corrected.

L24 "essential addition". This is not demonstrated in the m/s. While the new DEM is an interesting addition to the 6 discussed, it is not the most accurate so to claim it is essential is unjustified.

Accept, the '*essential*' have been deleted.

L27-28. See first comment above. In addition, the two references cited do not use altimetry DEMs for ice motion tracking or mass balance.

Agree and accept, we have revised this sentence to avoid the existing misleading information: 'Knowledge of the detailed surface topography in Antarctica is essential for human fieldwork, ice motion tracking (Bamber et al., 2000) and numerical modelling of ice sheet (Cornford et al., 2015).'

The original sentence is also listed below:

The references have been corrected:
Bamber J L, Vaughan D G, Joughin I. Widespread complex flow in the interior of the Antarctic ice sheet. Science, 2000, 287(5456): 1248-1250.
Cornford S L, Martin D F, Payne A J, et al. Century-scale simulations of the response of the West Antarctic Ice Sheet to a warming climate. The Cryosphere, 2015, 9(4): 1579-1600.

L30 "monitoring the topography". What do the authors mean? Do they mean measuring elevation change? See first comment.

We have revised this sentence to avoid the existing misleading information:
'Digital elevation models (DEMs) of Antarctica, for example, can be used for **presenting**  the topography of ice sheets and ice shelves and …'.

L51-52. KU-band penetration into snowpack is unknown. This statement is misleading at best. There is plenty of literature on Ku band penetration into snow. Further, radar penetration can be corrected for either empirically or theoretically using a waveform fitting approach (see e.g. [Davis, 1996; 1997]).

Agree and accept. We have revised this sentence to avoid the misleading information:
'Although the radar penetration depth of the Ku-band into snowpack can be corrected for either empirically or theoretically using a waveform fitting approach (Davis, 1996; Davis, 1997), the spatial and temporal variations of penetration depth are still difficult to account. As multi-temporal and large-scale satellite altimeter data are usually used, this includes some uncertainties in the elevation estimation.'.

The original sentence is also listed below:

References:
Davis C H. Temporal change in the extinction coefficient of snow on the Greenland ice sheet from an analysis of Seasat and Geosat altimeter data. IEEE transactions on geoscience and remote sensing, 1996, 34(5): 1066-1073.
Davis C H. A robust threshold retracking algorithm for measuring ice-sheet surface elevation change from satellite radar altimeters. IEEE Transactions on Geoscience and Remote Sensing, 1997, 35(4): 974-979.

L84 Icessn?

IceBridge ATM L2 Icessn elevation, slope and roughness (V002) product (Studinger et al., 2014) is used here for DEM evaluation. According to Studinger et al. (2014), '… *the fundamental form of ATM topography data is a sequence of laser footprint locations acquired in a swath along the aircraft flight track. The **icessn program** condenses the ATM surface elevation measurements by fitting a plane to blocks of points selected at regular intervals along track and several across track. …*'.
Here, Icessn is a terminology.

Reference:
Studinger, M.: IceBridge ATM L2 Icessn Elevation, Slope, and Roughness, version 2. Boulder, Colorado USA: National Snow and Ice Data Center, Digital media, https://doi.org/10.5067/CPRXXK3F39RV, 2014.

L87-88. This is not reducing "seasonal elevation changes" because any change that occurs in less than a year must be sub-annual, not multi-annual. Second, except for the pensinsula seasonal changes do not really exist over Antarctica, they are sub-annual which is not the same thing.

*Agree and accept, the 'seasonal changes' has been replaced by 'sub-annual changes' in the revised manuscript.*

Table 1, p5. It states the ICESat/ERS-1 DEM has an unclear timestamp but in the paper there is a specific section the DEM time stamp, where it explains that the ICESat data are corrected for any significant dh/dt between 1995 and date of acquisition [Bamber et al., 2009] so the time stamp is extremely clear and spelled out. It may not be perfectly corrected to 1995 but the Table is incorrect for this DEM and, consequently, makes me concerned about the veracity of claims made elsewhere in Table 1. Without carefully studying the methods used to generate each DEM the authors will not be able to support the claims made in Table 1.

*Agree and accept, all DEMs have specific time stamps, but the temporal resolutions are different. In order to avoid the misleading information in original Table 1, we delete the column [time stamp] as the column [Time span of applied source data] has provided the information for DEM time stamps. All the information in Table 1 have been checked, the updated Table 1 is also shown below:*

**Table 1.** Detailed introductions to six previously published Antarctic DEMs, including the source data, time span of the source data, spatial posting/resolution.

| DEM | Source data | Time span of applied source data | Spatial posting/resolution |
|---|---|---|---|
| ICESat DEM | ICESat | February 2003 to June 2005 | 500 m |
| ICESat/ERS-1 DEM | ICESat, ERS-1 | 1994-1995, 2003-2008 | 1 km |
| Slater CryoSat-2 DEM | CryoSat-2 | July 2010 to July 2016 | 1 km |
| Helm CryoSat-2 DEM | CryoSat-2 | A full 369-day-long cycle starting January 2012 | 1 km |
| REMA DEM | GeoEye-1, WorldView-1/2/3 | As of July 2017 | Variable resolutions, 2 and 8 m |
| TanDEM-X PolarDEM | TerraSAR-X, TanDEM-X | April to November 2013, April to October 2014, mid-2014, July 2016 to September 2017 | 90 m |

L120 replace "seasonal" with sub annual here and elsewhere.

*Accept, the related parts have been revised.*

L130 neighboring. The preferred spelling convetion for TC is UK English not US. Please follow that convention.

*This has been revised.*

L153. Strictly speaking the kriging variance is not, and cannot, be used to determine the interpolation error. It is related to the interpolation error but not equal to it.

Here, kriging variance error is not used to determine the interpolation error, but to calculate the elevation uncertainty due to the kriging interpolation (according to Slater et al. (2018)). The interpolation error is evaluated by using OIB data in Table 3.

Reference:
Slater T, Shepherd A, McMillan M, et al. A new digital elevation model of Antarctica derived from CryoSat-2 altimetry. The Cryosphere, 2018, 12(4): 1551-1562.

L171. Nominal resolution. See above. It would be more appropriate to refer to 250 m posting with a resolution of 1 km for latitudes x-y and 250 m from a-b…

We have corrected this misleading terminology, and revised the related parts in the manuscript.

L248. Needs rewording.

We have revised this sentence:
'Larger biases will be included in the ICESat-2 DEM if the coverage of interpolated elevations is high, hence the elevation gaps in the 500 m DEM are firstly filled by the resampled 1 km DEM to reduce the coverage of interpolated elevations.'.

The original sentence is also listed below:
'Larger biases are included in the ICESat-2 DEM if the coverage of interpolated elevations is high, which demonstrates the reasonability of the three resolutions used for DEM generation from ICESat-2.'

L250 Here the authors claim the bias increases with slope or roughness but that is not borne out by Table 3. See General Comments.

This has been responded in the General Comments (the second one).

L280-281. This is a sweeping generalisation that is incorrect. The accuracy of photogrammetric derived DEMs is a function of the resolution of the sensor and the accuracy of GCPs.

Agree and accept, we have revised this sentence:
'As shown in Table 5, REMA DEM and TanDEM PolarDEM are more accurate than altimeter-derived DEMs; hence, similar elevations indicate the reliability of ICESat-2 DEM in mountain environments.'.

The original sentence is also listed below:
'DEMs derived from stereo-photogrammetry have a better performance for high-slope regions (Slater et al., 2018); hence, similar elevations indicate the reliability of ICESat-2 DEMs in mountain environments.'

L321. "modal resolution" -> posting.

Done.

References

Bamber, J. L., Gomez Dans, J. L., and Griggs, J. A. (2009), A new 1 km digital elevation model of the Antarctic derived from combined satellite radar and laser data. Part I: Data and methods, The Cryosphere 3(2), 101-111.

Davis, C. H. (1996), Temporal change in the extinction coefficient of snow on the Greenland ice sheet from an analysis of seasat and geosat altimeter data, IEEE Trans. Geosci. Remote Sensing, 34(5), 1066-1073.

Davis, C. H. (1997), A robust threshold retracking algorithm for measuring ice-sheet surface elevation change from satellite radar altimeters, IEEE Trans. Geosci. Remote Sensing, 35(4), 974-979.

---

## Author Comment (AC2)

We thank the reviewer for the helpful feedback, these suggestions have significantly improved the text and figures, we are appreciative of the help and time.

We have addressed all the comments here, point by point responses to the comments are listed in BLUE.

Here we summarize two major revisions in the revised manuscript:

First, in the revised manuscript we select a 500m posting for the derived ICESat-2 DEM. That is, elevations are firstly estimated at the resolution of 500 m and 1 km, respectively. Then the observation gaps in the 500 m DEM are filled by the resampled 1 km DEM (resampled to the 500 m), and the remaining gaps are filled by using ordinary kriging interpolation. In the original manuscript, we firstly generated four DEMs, i.e., 250m DEM, 500m DEM, 1km DEM and interpolated DEM. We produced a DEM at 250 m posting by resampling DEMs at 500 m and 1 km posting (including the interpolated DEM). However, 250m DEM only has a coverage of 26%, most of the derived DEM were based on resampled coarser DEMs. Hence, as suggested in your comment below, starting from 500m, refill with 1km and do the kriging would make more sense. In this case the model fitting tends to be more robust as more points are used per grid cell. The updated Figure 4 is listed below (the figure indexes in this text are the same to these in original manuscript):

[Figure]

**Figure 4.** (a) A fine-scale DEM of Antarctica at a posting of 500 m derived from ICESat-2, which covers both the ice sheet and ice shelves with the southern limit of 88°S. (b) Map of the ICESat-2 DEM elevation uncertainty.

Second, in order to provide a robust and reasonable comparison between ICESat-2 DEM and other DEMs, a common OIB data in areas of low elevation change from 2009 to 2019 are used in the revised manuscript. As the same evaluation data are used, all the DEMs can be compared validly. These OIB data are located in the ice sheet interior, as shown in the figure below (Figure 1b). The CryoSay-2 Low Rate Mode (which was designed for flat ice sheet interior measurements) mask is used to extract the regions of low elevation change. CryoSat Geographical Mode Mask (v 4.0, updated in 19 August - 26 August 2019) at https://earth.esa.int/eogateway/news/cryosat-geographical-mode-mask-4-0-released is used here. The averaged elevation change rate in the used

OIB data locations is about -0.0074±0.0821 m/yr from 2003 to 2019, according to elevation change rate data from Smith et al. (2020). Hence, we assume that in these areas the effect of the elevation change on the DEM evaluation can be ignored. It should be noted that, when evaluating ICESat-2 DEM individually, OIB data in areas of low elevation change from 2009 to 2017 and OIB data in all Antarctica from 2018 to 2019 (Figure 1a) are used for a comprehensive evaluation. In addition, published GPS transects (Schröder et al., 2017) are also used here for the DEM comparison (Figure 1c).

[Figure]

Elevation (m)

0    500    1000    1500    2000    2500    3000    3500    4000

**Figure 1.** (a) Map of the OIB airborne data in October and November 2018 and October and November 2019 in Antarctica. (b) Map of the OIB airborne data from 2009 to 2019 in Antarctic ice sheet interior. (c) Map of the GPS transects from 2001 to 2015 in Antarctica. The dashed lines in (b) and (c) show the boundary of region where we assume to have low elevation change, it is the mode mask boundary of CryoSat-2 Low Rate Mode data in Antarctica, this data mode was designed for flat ice sheet interior measurements.

In addition, in the revised manuscript we resample all the DEMs to the OIB data locations and calculate the difference and statistics to perform the evaluation. The original approach (calculating a median OIB elevation for each grid cell) will certainly influence the evaluation results as the DEMs have different pixel spacing. Also as pointed out in your comment below, OIB is the reference elevation and cannot be replaced by the median values. The new approach is also applied for DEM evaluation using GPS data. The related statistics have changed, but the same conclusions are found comparing to original statistics. The updated Tables 3, 5 and 6 are listed in below:

**Table 3.** Comparisons between the ICESat-2 DEM and OIB airborne elevation measurements (including data in areas of low elevation change from 2009 to 2017 and data in all Antarctica from 2018 to 2019) in observed and interpolated areas for individual regions (i.e., the ice sheet and ice shelves).

|  | Region | MeD (m) | MeAD (m) | SD (m) | RMSD (m) | Number of used OIB measurement points |
|---|---|---|---|---|---|---|
| Observed | Ice sheet | 0.08 | 1.18 | 12.75 | 12.75 | 3589087 |
|  | Ice shelves | 0.77 | 2.60 | 15.26 | 15.27 | 191754 |
|  | Total | 0.09 | 1.23 | 12.89 | 12.89 | 3780841 |
| Interpolated | Ice sheet | -0.40 | 2.50 | 20.68 | 20.73 | 1237416 |
|  | Ice shelves | 0.36 | 3.23 | 24.61 | 24.65 | 185613 |

|  |  | -0.33 | 2.58 | 21.25 | 21.28 | 1423029 |
|---|---|---|---|---|---|---|
|  | Total | | | | | |
| Overall | Ice sheet | 0.01 | 1.41 | 15.20 | 15.20 | 4826503 |
|  | Ice shelves | 0.59 | 2.88 | 20.40 | 20.43 | 377367 |
|  | Total | 0.03 | 1.49 | 15.64 | 15.64 | 5203870 |

**Table 5.** Comparisons between the ICESat-2 DEM, ICESat DEM, ICESat/ERS-1 DEM, Helm CryoSat-2 DEM, Slater CryoSat-2 DEM, REMA DEM, TanDEM PolarDEM and OIB airborne elevation measurements in areas of low elevation change from 2009 to 2019.

|  | MeD (m) | MeAD (m) | SD (m) | RMSD (m) | Number of used OIB measurement points |
|---|---|---|---|---|---|
| ICESat-2 DEM | 0.10 | 0.98 | 5.36 | 5.38 |  |
| ICESat DEM | -2.61 | 6.35 | 19.90 | 20.43 |  |
| ICESat/ERS-1 DEM | -0.15 | 1.84 | 11.53 | 11.54 |  |
| Helm CryoSat-2 DEM | 0.65 | 2.68 | 24.97 | 25.02 | 1965309 |
| Slater CryoSat-2 DEM | 1.22 | 2.87 | 23.85 | 24.14 |  |
| REMA DEM | -0.16 | 0.53 | 1.75 | 1.76 |  |
| TanDEM PolarDEM | -2.84 | 2.94 | 2.76 | 3.90 |  |

**Table 6.** Comparisons between the ICESat-2 DEM, ICESat DEM, ICESat/ERS-1 DEM, Helm CryoSat-2 DEM, Slater CryoSat-2 DEM, REMA DEM, TanDEM PolarDEM and GPS elevation data in areas of low elevation change from 2001 to 2015.

|  | MeD (m) | MeAD (m) | SD (m) | RMSD (m) | Number of used OIB measurement points |
|---|---|---|---|---|---|
| ICESat-2 DEM | -0.03 | 0.41 | 1.17 | 1.17 |  |
| ICESat DEM | -1.91 | 2.89 | 5.21 | 5.97 |  |
| ICESat/ERS-1 DEM | -0.74 | 0.84 | 1.39 | 1.61 |  |
| Helm CryoSat-2 DEM | 0.07 | 0.67 | 1.67 | 1.71 | 488963 |
| Slater CryoSat-2 DEM | 0.00 | 0.46 | 1.65 | 1.66 |  |
| REMA DEM | 0.03 | 0.26 | 0.57 | 0.57 |  |
| TanDEM PolarDEM | -4.62 | 4.62 | 1.33 | 4.72 |  |

References:

Smith B, Fricker H A, Gardner A S, et al. Pervasive ice sheet mass loss reflects competing ocean and atmosphere processes. Science, 2020, 368(6496): 1239-1242.

Schröder L, Richter A, Fedorov D V, et al. Validation of satellite altimetry by kinematic GNSS in central East Antarctica. The Cryosphere, 2017, 11(3): 1111-1130.

Review:

A fine-scale digital elevation model of Antarctica derived from ICESat-2

Shen et.al. 2021

The study presents a new elevation model of Antarctica based on one-year of ICESat-2 observations. The authors provide a specific time-stamped DEM with a final pixel size of 250m, following the same approach as presented by Slater et.al. (2017).

The new DEM is validated against OIB data and compared to existing Antarctic DEMs. Results show an improved accuracy compared to DEMs based on Radar altimetry but with less accuracy than DEMs based on Radar interferometry or Stereo-Photogrammetry.

In general, it is an interesting project and worth to be published as ICESat2 provides precise point information with high accuracy and good coverage. This large data base should be used to generate a gridded data product of high quality which is easily accessible and to be used in different applications. The authors did this approach; however, I do have some concerns and questions related to the method, the validation and comparison with existing DEMs.

Thank you very much for your feedback and advice, these suggestions have significantly improved the text and figures, we are appreciative of your comments.

Generell comments:

The paper is well written, however in some instances the statements are not accurate (see below). Figures are mostly ok with room for improvements (see below).

We have revised the related parts of the manuscript according to your comments, details are listed in below.

Structure is fine and easy to follow.

I have some concerns about the selected 250m posting and the fitting method used for 1 year of data.

1. As we mentioned in the very beginning, we produce a new ICESat-2 DEM at 500 m posting, the 500m DEM is refilled with estimated and interloped DEMs at 1 km posting to give a more reasonable and robust elevation estimate.

2. The map of elevation change rate in this study is shown in below, we also provide the estimation result from 2003 to 2019 in Smith et al. (2019) for a comparison. Overall, similar elevation change patterns can be found between the two figures. For example, larger elevation decreases can be found in the margin of West Antarctica, obvious elevation increases can be found in the interior of West Antarctica (red cycles in the figure). Hence, the elevation change pattern based on one-year ICESat-2 data is reasonable, which indicates that one year of data can give a reliable elevation change map and the elevation estimation is thus reliable. This may due to the much higher measurements density and accuracy of ICESat-2 than previous altimeters. In addition, ICESat-2 DEM has a higher accuracy than other altimeter-derived DEMs by comparing to both airborne and GPS data, which also proves the feasibility of the data and method.

[Figure]

**Figure.** Map of elevation change rate in Antarctica derived from one year of ICESat-2 data in this study (left) and map of elevation change rate in Antarctica from 2003 to 2019 in Smith et al. (2019) (right).

Reference:

Smith B, Fricker H A, Gardner A S, et al. Pervasive ice sheet mass loss reflects competing ocean and atmosphere processes. Science, 2020, 368(6496): 1239-1242.

Major question marks arise when looking at figure 3b illustrating the difference between the three-postings used to generate the final DEM.

In the Fig. 3, elevations at different latitudes are presented by the region-averaged values. As shown in the Fig. 2 (in below), the coverages of DEMs at three resolutions are different, the observed elevation grid cells at the same region are not completely overlapped, DEM at coarser resolution has more observed grid cells than DEM at fine resolution at the same region, hence the region-averaged elevation values have large differences.

[Figure]

**Figure 2.** Map of the observed grid cells of DEMs at the spatial resolution of 250 m (a), 500 m (b) and 1 km (c). The observed grid cells are colored in blue, the overall coverage of each DEM in Antarctica is also presented beside.

In the revised manuscript, we compare the elevations at two resolutions (i.e., 500 m and 1 km, as we mentioned in the very beginning only two resolutions are used in the revised manuscript) in the overlapped regions in Antarctica. The map and histogram of elevation difference between 1km DEM and 500m DEM are shown in Figs. 3b and 3c. An averaged elevation difference of $0.04 \pm 2.93$ m is found, which is quite small comparing to the estimated elevations. This elevation difference is acceptable and it is valid to combine elevations at these two resolutions.

[Figure]

**Figure 3.** (a) Spatial coverages of observed grid cells in the five latitude ranges when two spatial resolutions, i.e., 500 m (blue) and 1 km (red), are applied. (b) Map of the elevation difference of DEMs at the resolutions of 1 km and 500 m. (c) Histograms of the elevation difference of DEMs at the resolutions of 1 km and 500 m, the average and standard deviation values are also presented.

I have some major concerns about the method used to validate the new DEM and the way to compare to existing DEMs.

As we mentioned in the very beginning, a new evaluation method (i.e., we resample the DEM to the OIB data locations and calculate the difference and statistics) has been applied in the revised manuscript. A common OIB data in areas of low elevation change from 2009 to 2019 are used for a robust and reasonable comparison between ICESat-2 DEM and other DEMs. GPS data are also used here for the comparison, the same evaluation method is applied.

Specific comments and questions

Section 2.1

Please state which processing version was used. Do you apply any additional filtering prior fitting the data than the atl06_quality_summary?

ALT06 Version 3 data are used in this study. No additional filtering prior fitting the data than the atl06_quality_summary is applied. These have been added into the revised manuscript.

The spatial resolution of 20m is true for the along track sampling. However, the track spacing is

latitude dependent and I doubt that a 250m spatial resolution across track is reached for lower latitudes. Please be correct here, and add a figure showing the latitude dependency of the track spacing. This is important as one of your arguments is the dense spatial coverage and I would like to know if 250m is a reliable grid size. As seen later in the manuscript most of the grid cells used are of coarser resolution.

Agree. The Advanced Topographic Laser Altimeter System (ATLAS) onboard ICESat-2 has six beams (three pairs), which can provide a large measurement density (Neumann et al. 2019). The three pairs of beams are separated by 3.3 km across tracks, and two beams in one pair are separated by 90 m across the track. In addition, prior to April 1 2019, ICESat-2 data collected during that time was not along planned reference ground tracks (RGTs). Hence, the ICESat-2 ground tracks have complexed spatial distributions, the latitude dependency of the track spacing is difficult to estimate.

Instead, we show the spatial coverages of DEMs at 250 m, 500 m and 1 km (Fig. 2) here, which is much easier to evaluate if a 250m (500m) DEM posting makes sense. In lower latitudes there are still some grid cells with estimated elevations. This means the 250m spatial resolution across track can be reached for lower latitudes. However, as we mentioned in the very beginning, 250m DEM only has 26% coverage in original DEM, most of the derived DEM were based on resampled coarser DEMs and interpolation. Hence, in the revised manuscript we produce a new ICESat-2 DEM at 500 m posting. 500m is a more reliable grid size which makes denser spatial coverage of the observed elevations.

[Figure]

**Figure 2.** Map of the observed grid cells of DEMs at the spatial resolution of 250 m (a), 500 m (b) and 1 km (c). The observed grid cells are colored in blue, the overall coverage of each DEM in Antarctica is also presented beside.

Reference:
Neumann T A, Martino A J, Markus T, et al. The Ice, Cloud, and Land Elevation Satellite–2 Mission: A global geolocated photon product derived from the advanced topographic laser altimeter system. Remote sensing of environment, 2019, 233: 111325.

Section 2.2

L84 ATM L2 Icessn?

IceBridge ATM L2 Icessn elevation, slope and roughness (V002) product (Studinger et al., 2014) is used here for DEM evaluation. According to Studinger et al. (2014), '... *the fundamental form of ATM topography data is a sequence of laser footprint locations acquired in a swath along the aircraft flight track. The* **icessn program** *condenses the ATM surface elevation measurements by fitting a plane to blocks of points selected at regular intervals along track and several across track. ...*'.

Here, Icessn is a terminology.

Reference:

Studinger, M.: IceBridge ATM L2 Icessn Elevation, Slope, and Roughness, version 2. Boulder, Colorado USA: National Snow and Ice Data Center, Digital media, https://doi.org/ 10.5067/CPRXXK3F39RV, 2014.

OIB Slope and roughness is mentioned two times and plotted in Fig 1. I don't see where this information is used in the paper and how the histograms add information. I suggest to remove b,c,d in the Figure. In the text it is also not clear what filtering is exactly applied (Line 85).

Labels in Fig 1 b,c,d are too small.

Accept, Figs. 1b, c and d have been removed. We add the related references about the filter method here:

'… and a data filter (Young et al., 2008; Kwok et al., 2012; Studinger et al., 2014) is applied to remove abnormal values due to geolocation errors or cloud cover.'

References:

Young D A, Kempf S D, Blankenship D D, et al. New airborne laser altimetry over the Thwaites Glacier catchment, West Antarctica. Geochemistry, Geophysics, Geosystems, 2008, 9(6).

Kwok R, Cunningham G F, Manizade S S, et al. Arctic sea ice freeboard from IceBridge acquisitions in 2009: Estimates and comparisons with ICESat. Journal of Geophysical Research: Oceans, 2012, 117(C2).

Studinger, M.: IceBridge ATM L2 Icessn Elevation, Slope, and Roughness, version 2. Boulder, Colorado USA: National Snow and Ice Data Center, Digital media, https://doi.org/ 10.5067/CPRXXK3F39RV, 2014.

Section 2.3

It is not correct that the Altimetric based DEMs have no specific time stamp. Slater DEM corrects for elevation change and though the DEM effective time is 1st July 2013. The Helm DEM is July 1st 2012. One cycle (369 days) was used to generate this DEM. IceSat/ERS1 DEM has also a clear timestamp as elevation change was taken into account.

Agree and accept, all DEMs have specific time stamps, but the temporal resolutions are different. In order to avoid the misleading information in original Table 1, we delete the column [time stamp] as the column [Time span of applied source data] has provided the information for DEM time stamps.

All the information in Table 1 have been checked, the updated Table 1 is also shown below:

**Table 1.** Detailed introductions to six previously published Antarctic DEMs, including the source data, time span of the source data, spatial posting/resolution.

| DEM | Source data | Time span of applied source data | Spatial posting/resolution |
|---|---|---|---|
| ICESat DEM | ICESat | February 2003 to June 2005 | 500 m |
| ICESat/ERS-1 DEM | ICESat, ERS-1 | 1994-1995, 2003-2008 | 1 km |
| Slater CryoSat-2 DEM | CryoSat-2 | July 2010 to July 2016 | 1 km |
| Helm CryoSat-2 DEM | CryoSat-2 | A full 369-day-long cycle starting January 2012 | 1 km |
| REMA DEM | GeoEye-1, WorldView-1/2/3 | As of July 2017 | Variable resolutions, 2 and 8 m |
| TanDEM-X PolarDEM | TerraSAR-X, TanDEM-X | April to November 2013, April to October 2014, mid-2014, July 2016 to September 2017 | 90 m |

This means one of your arguments doesn't hold and you need to check the whole paper where this misleading information is given (it is stated already in the abstract).

The related parts about the DEM time stamps have been revised, we have removed all the misleading statements (i.e., altimetric based DEMs have no specific time stamp) in the revised manuscript.

Additional in Table1: REMA and TDX DEM: Instead of marking time stamp unclear you should give the acquisition time of the sensors. If I remember correctly TDX was in Antarctic acquisition mode for two seasons (2013/2014 and to fill gaps 2016/2017). REMA used data mostly from the 2015 and 2016 austral summer seasons. Please carefully check the relevant publications of Wessel (2021) and Howat (2019).

All the information in the updated Table 1 have been checked.

Section 2.4

L128 elevation gasps

Revised. It should be elevation gaps.

I don't understand the labeling of Figure 2 (The numbers of ICESat-2 measurement points in grid cells at 250 m and 500 m are both resampled to a resolution of 1 km.). Is this figure showing the 1km coverage or not? The final DEM is 250m so why do you show the 1km coverage?

The original figure shows the 1km coverage. This figure is updated (as shown in below) according

to your suggestion. As we generate a new DEM at 500m posting, we give the map of the numbers of valid ICESat-2 measurement points in each 500 m grid cell in the new figure.

[Figure]

**Figure.** Map of the numbers of valid ICESat-2 measurement points in each 500 m grid cell. The numbers of ICESat-2 measurement points in 1 km grid cells are resampled to the resolution of 500 m.

Why don't you show the coverage for all three resolutions (e.g. panels a), b), c)) and include in the figure label how many of the grid cells have data coverage (e.g 1km 74 %, 500m 46%, 250m 26%).

Accept. Here we add a new figure to show the coverages of DEMs at the resolution of 250 m, 500 m and 1 km in the revised manuscript (Fig. 2).

[Figure]

**Figure 2.** Map of the observed grid cells of DEMs at the spatial resolution of 250 m (a), 500 m (b) and 1 km (c). The observed grid cells are colored in blue, the overall coverage of each DEM in Antarctica is also presented beside.

Doing so it would be much easier to evaluate if a 250m DEM posting makes sense. In your case you only have 26% coverage, so most of the 250m DEM is based on resampled coarser DEMs and interpolation. Therefore I think, starting from 500m, refill with 1km and do the kriging would make more sense. Later one can resample to whatever posting is needed. Doing so, your fitting should be more robust as more points are used per pixel. As comparison, Slater stated a 60% coverage for his finest 1km grid, which seems to me more reasonable.

Agree and accept. As we mentioned in the very beginning, we produce a new ICESat-2 DEM at 500 m posting, the 500m DEM is refilled with estimated and interloped DEMs at 1 km posting to give a more reasonable and robust elevation estimate.

In addition, I have some concerns if one year of data is enough to estimate a reliable elevation change which is internally used to reference your DEM to a specific time stamp. For a six-year time series as in Slater et.al. this makes sense but for one year I don't see the point. Could you please provide the elevation change product (a5 parameter of your fit) to see if the method makes sense?

Otherwise, one could remove a5 from equation 1 and make the fit more robust.

This has been responded in the General Comments (the first one).

Could you please explain why you chose the fitting method? To my opinion the fitting method forces a quadratic surface in each grid cell (but mostly the real surface is not quadratic – sastrugis, small scale undulations etc.). The fitting method is minimizing the advantage of ICESat2. As the accuracy of each single IceSat-2 measurement is very high and the footpring small the quality of the input data is very high (compared to Radar altimetry). So why not make use of all valid measurements by taking all data and run the kriging interpolation (or whatever you prefer) – similar to the Helm or Bamber DEM approach?

A model fitting method used here is to separate the various contributions to the estimated elevations within each grid cell (Flament and Remy, 2012; McMillan et al., 2014), including local surface terrain and elevation change. This function is fitted in each grid cell by using an iterative least-squares fit to all the elevation measurements to minimize the impact of outliers. A quality control criterion is also used to reduce the effect of any poor fit. This method suits ICESat-2 orbit cycle, which samples dense ground tracks comparing to previous satellite radar altimeters, more measurement points are included in the grid cell and the estimated elevations are more robust. The resolutions of grid cells (i.e., 500 m and 1 km) are appropriate for the used ICESat-2 data in this study. First, most elevations (72%) can be directly estimated based on this method. Second, it is possible for a quadratic form to model the topography at these scales and smaller elevation residuals can be found than using a simple linear fit (Flament and Remy, 2012).

Approximately $4.69 \times 10^9$ ICESat-2 measurement points are used for elevation estimation in this study, which has a coverage of 18% for the Antarctica. The direct application of kriging interpolation based on all valid measurements means the 72% elevations are estimated from interpolation. As the evaluation results shown in this study and Slater et al. (2018), the bias of observed elevations is obviously smaller than that of interpolated elevations, hence the interpolation ratio should be reduced as possible. The model fitting method considers the various contributions to the estimated elevations by including all data acquired within each region, the interpolation ratio is reduced and the derived elevations can represent the elevation in each region well. In addition, model fitting method can provide the estimation of elevation change rate, and the estimate agrees well with accurate elevation change estimations from crossover-point method (Moholdt et al., 2010), which

provides an addition reference for the research of ice dynamics and mass balance.

Besides, Kriging interpolation based on $4.69 \times 10^9$ ICESat-2 measurement points is quite time-consuming now.

The above discussion has also been added into the manuscript.

References:

Moholdt G, Nuth C, Hagen J O, et al. Recent elevation changes of Svalbard glaciers derived from ICESat laser altimetry. Remote Sensing of Environment, 2010, 114(11): 2756-2767.

Flament T, Rémy F. Dynamic thinning of Antarctic glaciers from along-track repeat radar altimetry. Journal of Glaciology, 2012, 58(211): 830-840.

McMillan M, Shepherd A, Sundal A, et al. Increased ice losses from Antarctica detected by CryoSat-2. Geophysical Research Letters, 2014, 41(11): 3899-3905.

Slater T, Shepherd A, McMillan M, et al. A new digital elevation model of Antarctica derived from CryoSat-2 altimetry. The Cryosphere, 2018, 12(4): 1551-1562.

Could you please spent more explanation or any equation of how your uncertainty map was derived. What is the 95% confidence level for elevation estimation and how exactly is this derived?

The elevation uncertainty of observed grid cell is derived based on the fitting performance, which is estimated from the equation below:

$$u = t(1\text{-}0.025, n\text{-}p) \times SD$$

where u is the uncertainty of elevation estimate, SD is the standard error of the elevation estimate, and $t(1\text{-}0.025, n\text{-}p)$ is the 95% percentile of $t$-distribution with $n\text{-}p$ degrees of freedom, $n$ is the number of ICESat-2 measurement points in the grid cell, $p$ is the number of regression coefficients (i.e., 7). This is calculated based on MATLAB R2017a.

For the interpolated grid cells, elevation uncertainty is calculated from the kriging variance error (as shown in the comment below), ESRI ArcGIS 10.6 is used here.

The above statement has also been added into the manuscript.

What kriging method did you apply (model, nugget, sill, radius). How is the variance error calculated. Which software was used for kriging.

Ordinary kriging is used (Semi-variogram model: spherical, nugget: 0, sill: 1652285.953, radius: 10 km). ESRI ArcGIS 10.6 is used for kriging and kriging variance error.

According to ArcGIS Desktop Help, the kriging variance error contains the kriging variance at each output raster cell. Assuming the kriging errors are normally distributed, there is a 95.5 percent probability that the actual z-value (i.e., elevation) at the cell is the predicted raster value ± two times

the square root of the value in the variance error.

Fig3: I don't understand the large differences between the DEMs of different resolution. How can a 100-300m offset be explained? Two options: Your method isn't working or the evaluation as shown in fig3b makes no sense. It would be better to show Antarctic wide difference plots of (DEM_250m – resampled DEM_500m and DEM_250 – resampled DEM_1km) and the corresponding histogram and statistics.

This has been responded in the General Comments (the second one).

Section 2.4.3

Why don't you resample the DEM to the OIB data locations and calculate the difference and its statistics? OIB is your reference elevation and you shouldn't replace is by a median. By calculating a median for each grid cell, you assume the surface in the grid cell is flat. In the interpolation you assumed a quadratic surface.

Agree and accept. In the revised manuscript we resample all DEMs to the OIB data locations and calculate the difference and statistics. This has been responded in the very beginning.

Table3: Why are ice shelves less accurate? Do you have any explanation? These are very flat areas and your argument that the DEM accuracy is better in flat areas doesn't hold anymore. Did you apply a tide correction for the OIB data?

As we mentioned in the very beginning, we generate a new DEM at 500 m posting, perform a new evaluation by resampling the DEM into OIB location and calculating the statistics. According to the updated evaluation result, we can still find a slight better performance of DEM in ice sheet than ice shelves (a tide correction has been applied for the OIB data). In order to find the explanation, we present the histograms of surface slope and roughness values (derived from OIB data) for ice sheet and ice shelves in below:

[Figure]

**Figure.** Histograms of the OIB-derived surface slope and roughness values for ice sheet and ice

shelves.

As we can found in this figure, observed ice shelves have overall smaller surface roughness than ice sheet, but have a larger percentage of high-slope areas than ice sheet. For example, approximately 70% of the OIB measurement points which covered ice sheet have slope values of < 0.01°. In comparison, approximately 50% of the OIB measurement points which located in ice shelves have slope values of < 0.01°. Hence, observed ice shelves have a higher percentage of high-slope areas, which may cause larger elevation biases. To test this argument, standardized regression coefficients between surface slope/roughness and the elevation difference (i.e., mean absolute difference between ICESat-2 DEM and OIB elevations) are calculated here by using a multivariate linear regression model (this model is fitted by using an iterative least-squares fit). All OIB data in 2018 and 2019 are used. Standardized values of surface slope, roughness and elevation difference are used for a valid comparison. The regression coefficients for surface slope and roughness are 0.18 and -0.01. Larger regression coefficient indicates that the surface slope has greater effect on elevation difference than roughness. Hence, although ice shelves observed by OIB data have smaller surface roughness than ice sheet, a higher percentage of high-slope areas makes ice shelves have a slight worse DEM performance.

In addition, similar phenomenon can still be found in Table 2 in Slater et al. (2018), their observed DEM also has a slight better performance in ice sheet than ice shelves.

**Table 2.** Statistics of the comparison between observed and interpolated DEM grid cells and airborne elevation measurements for individual Antarctic regions and mode mask areas. In total, only 5 and 2 % of DEM elevation values are obtained through interpolation for the ice sheet and ice shelves, respectively.

| Region | Observed | | | Interpolated | | |
|---|---|---|---|---|---|---|
| | Number of compared grid cells | Median difference (m) | rms difference (m) | Number of compared grid cells | Median difference (m) | rms difference (m) |
| Ice sheet | 230 165 | −0.27 | 13.36 | 32 933 | 25.37 | 138.62 |
| Ice shelves | 40 081 | −0.42 | 14.31 | 4772 | 1.20 | 30.96 |
| Antarctic Peninsula | 6820 | −1.12 | 22.40 | 7473 | 82.21 | 191.07 |
| West Antarctica | 60 452 | −0.86 | 11.43 | 8783 | 11.78 | 96.15 |
| East Antarctica | 162 893 | −0.17 | 13.60 | 14 679 | 19.62 | 117.77 |
| LRM | 73 867 | 0.26 | 7.15 | 1683 | 6.51 | 41.70 |
| SARIn (ice sheet only) | 156 298 | −0.82 | 15.45 | 31 250 | 28.65 | 141.97 |
| Total | 270 246 | −0.30 | 13.50 | 37 655 | 19.84 | 131.13 |

Reference:

Slater T, Shepherd A, McMillan M, et al. A new digital elevation model of Antarctica derived from CryoSat-2 altimetry. The Cryosphere, 2018, 12(4): 1551-1562.

Section 3

Figure 6b and 6d. It seems that the DEM has artefacts, especially over the ice shelf. What is the reason for those artefacts? (Fitting routine or kriging artefacts or IceSat2 data problems?).

Do you observe similar artefacts in other areas as well? Could you please zoom into 6b and compare this region to hill shades of the other DEMs?

Yes, we update the ice boundary for Larsen C ice shelf in this study. In July 2017, a giant tabular iceberg (named A-68) calved from Antarctica's Larsen C ice shelf. It is necessary to remove this region from our DEM as our data is from 2018 to 2019. The ice boundary was manually extracted based on the MODIS image (250 m resolution), no changes are made in other regions. This statement has also been added into the manuscript.

What is the reason for plotting the grounding line? To me it is an unnecessary information.

Accept. The grounding line has been removed from the figure.

Figure 7c: too small and low resolution. The color of 500m and 1km grid cells cannot be distinguished (I'm red green blind). Colors of 7a and 7c should be the same.

The resolutions of all figures in the manuscript have been improved, color green is not used in the updated figure. The colors of 7a and 7c are the same in the updated figure.

[Figure]

**Figure 7.** (a) Coverages of observed grid cells at 500 m and 1 km and interpolated grid cells in 27 drainage basins of ice sheets (Zwally et al., 2012) and ice shelves. The boundaries and basin index (ID) of 27 ice sheet drainage basins (Numbers 1 to 27) and ice shelves (Number 28) are shown in (b). The coverages of observed (at two spatial resolutions) and interpolated grid cells in the Antarctic Peninsula, West Antarctica, East Antarctica and ice shelves are also shown in (b). (c) Map of the selected grid cell resolution for deriving the ICESat-2 DEM in all grid cells at a spatial resolution of 500 m. Elevation values derived from 1 km and interpolation (i.e., 1 km) are resampled to a resolution of 500 m.

Figure 8: Please chose a different color scale (e.g., red-white-blue like in Figure 9)

Accept. Red-white-blue like in Figure 9 has been applied in this figure.

[Figure]

**Figure 8.** (a) Map of the difference between the ICESat-2 DEM and OIB airborne elevation measurements. Detailed maps of the ice sheet interior, Pine Island Glacier region and ice sheet margin in East Antarctica are shown in (b), (c) and (d), and their locations are also shown in (a) by red rectangular boxes. The background is the shaded relief map of Antarctica derived from the ICESat-2 DEM.

Section 4

Comparison to other DEMs and DEM comparison to OIB data is difficult to evaluate. The problem is the different time stamp of the DEMs. As you use for each DEM different OIB data which are not in the same area one can't compare the results. The table shows clearly different numbers of grid cells, so the results can't be compared.

This means that your arguments that the new DEM shows a better accuracy cannot derived from the applied analysis (I don't doubt that this is not the case but the analysis is inadequate to show this.)

Furthermore, your suggested approach to calculate a median OIB elevation for each grid cell will certainly influence the results as the DEMs have different pixel spacing (again resample the DEM to the OIB location).

A valid approach would be to choose OIB data in areas of low elevation change. This would enable you to take the whole OIB data set and compare the chosen data to all DEMs.

A similar comparison you applied in Table 6. However, the number of grid cells are not the same. Again, you can't compare the results.

In addition, for comparison one could use published GPS transects like:

Brunt, K. M., Neumann, T. A., and Larsen, C. F.: Assessment of altimetry using ground-based GPS data from the 88S Traverse, Antarctica, in support of ICESat-2, The Cryosphere, 13, 579–590, https://doi.org/10.5194/tc-13-579-2019, 2019.

Schröder, L., Richter, A., Fedorov, D. V., Eberlein, L., Brovkov, E. V., Popov, S. V., Knöfel, C., Horwath, M., Dietrich, R., Matveev, A. Y., Scheinert, M., and Lukin, V. V.: Validation of satellite altimetry by kinematic GNSS in central East Antarctica, The Cryosphere, 11, 1111–1130, https://doi.org/10.5194/tc-11-1111-2017, 2017.

Accept. These comments have been responded in the very beginning.

In addition, the same OIB data were used in the original Table 6. As the different grid cell resolutions of listed DEMs, the final grid cells used for comparison are different. In the revised manuscript this table has been removed as we used a common OIB data and GPS data for comparisons.

It would be also worth to show an OIB elevation profile and the differences of the different DEMs to this OIB profile (similar to Fig 9 in Slater et.al.) This could nicely show the performance of the new ICESAT-2 DEM.

Accept. In order to evaluate the DEM performance in more detail, the elevations along two OIB tracks in flat ice sheet interior and rough ice sheet margins are shown here. In the ice sheet interior where surface slopes are low, elevation differences of approximately 5 m can be found (the averaged elevation differences for ICESat-2 DEM, ICESat/ERS-1 DEM, Slater CryoSat-2 DEM, Helm CryoSat-2 DEM, REMA DEM and TanDEM-X PolarDEM are 0.03±1.01 m, 49.46±28.53 m, 0.02±4.16 m, -0.06±4.52 m, 0.20±2.17 m and -4.12±1.09 m). The elevation differences are further reduced when surface slope become smaller. While at the Pine Island Glacier where surface slopes are large, elevation differences of approximately 20 m can be found in the undulated terrains (the averaged elevation differences for ICESat-2 DEM, ICESat DEM, ICESat/ERS-1 DEM, Slater CryoSat-2 DEM, Helm CryoSat-2 DEM, REMA DEM and TanDEM-X PolarDEM are -0.40±19.43 m, 1.92±27.28 m, 1.24±14.20 m, 0.09±15.34 m, 2.69±13.67 m, 0.32±1.10 m and -0.99±0.92 m). Overall, ICESat-2 DEM has better performances in the flat regions than steep areas. Regions of low surface slope represent the majority of Antarctic ice sheet, hence most elevations from ICESat-2 DEM have smaller elevation biases.

[Figure]

**Figure.** Differences between the ICESat-2 DEM and OIB elevations along two OIB flight paths in (a) ice sheet interior and (b) Pine Island Glacier. ICESat-2 DEM elevations are in red, OIB elevations are in black, and the elevation differences between ICESat-2 DEM and OIB elevations are in blue. Locations of the two OIB flight paths are shown in red in the inserted figures of Antarctica.

Resolution of e.g. Fig 9 is too low

The resolution of all figures has been improved.

[Figure]

Figure 9. Elevation differences between the ICESat-2 DEM and six previously published DEMs, i.e., ICESat DEM (a), ICESat/ERS-1 DEM (b), Helm CryoSat-2 DEM (c), Slater CryoSat-2 DEM (d), REMA DEM (e) and TanDEM PolarDEM (f).

Section 5

I miss a clear statement, why this DEM is needed. REMA and TDX seem to outperform the new ICESat2-DEM by a factor of 2.

Here we discuss this issue in Section 5 in the revised manuscript:

Generally, the accuracy of ICESat-2 DEM is comparable to these of REMA DEM and TanDEM PolarDEM, which have higher resolutions and accuracies. However, these commercial images (used in REMA DEM and TanDEM PolarDEM) are expensive, significant economic investments are required to derive a large-scale DEM (e.g., for Antarctica) as a large amount of data are needed. More importantly, in this case multi-temporal images are usually used, the derived DEMs hence have various time stamps in different regions, which may cause some uncertainties in their further applications, e.g., numerical ice sheet modelling. ICESat-2 DEM has the same time stamp for all regions. Although the derived ICESat-2 DEM is less accurate than REMA DEM and TanDEM PolarDEM, considering the measurement accuracy of altimetry, these differences are still acceptable. Elevation change rate can be obtained when deriving the ICESat-2 DEM, which can provide an additional reference for ice topography and mass balance estimation. Comparing to altimeter-derived DEMs, ICESat-2 DEM has better (or comparable) performance in accuracy, resolution and coverage.

In addition, ICESat-2 data are easily accessible by the public comparing to high-resolution commercial images. In previous studies, several years of altimeter data are needed to derive the DEM in Antarctica. Due to the high-density measurements of ICESat-2, 13 months of ICESat-2 data can be used to generate a DEM for Antarctica, and the elevation accuracy is superior than other altimeter-derived DEMs. This means that the ICESat-2 DEM can be updated annually, this is still difficult for those DEMs derived from stereo-photogrammetry and interferometry. This study demonstrates the feasibility and reliability of using one-year ICESat-2 data to derive the Antarctic DEM, provides a reference for the processing scheme of DEM (e.g., in higher resolution, regularly updated) based on ICESat-2 in future.

Furthermore your uncertainty map shows values of < 1m, however the standard deviation of the differences to OIB shows 8m indicating that the uncertainty map is not representing this. Please discuss this.

Regions of lower surface slope which represent the majority of the Antarctic ice sheet – falls typically in the elevation uncertainty range < 1 m, hence our uncertainty map has more uncertainty values < 1m (67%). Large uncertainty values (i.e., > 10 m) can be found in the ice sheet margins or interpolated grid cells. Most of the OIB airborne data were obtained in the ice sheet margins or mountain environments of higher slope, where uncertainty values of ICESat-2 DEM and OIB

elevations are both larger. The ICESat-2 DEM and OIB elevation uncertainties, and the elevation error of ICESat-2 DEM, all affect the elevation differences between ICESat-2 DEM and OIB data, hence larger standard deviation of the differences can be found comparing to the uncertainty values (of < 1m) in the majority of the Antarctic ice sheet.

---

## Author Comment (AC3)

We thank the reviewer for the helpful feedback, these suggestions have significantly improved the text and figures, we are appreciative of the help and time.

We have addressed all the comments here, point by point responses to the comments are listed in BLUE.

Here we summarize two major revisions in the revised manuscript:

First, in the revised manuscript we select a 500m posting for the derived ICESat-2 DEM. That is, elevations are firstly estimated at the resolution of 500 m and 1 km, respectively. Then the observation gaps in the 500 m DEM are filled by the resampled 1 km DEM (resampled to the 500 m), and the remaining gaps are filled by using ordinary kriging interpolation. In the original manuscript, we firstly generated four DEMs, i.e., 250m DEM, 500m DEM, 1km DEM and interpolated DEM. We produced a DEM at 250 m posting by resampling DEMs at 500 m and 1 km posting (including the interpolated DEM). However, 250m DEM only has a coverage of 26%, most of the derived DEM were based on resampled coarser DEMs. Hence, as suggested by reviewers, starting from 500m, refill with 1km and do the kriging would make more sense. In this case the model fitting tends to be more robust as more points are used per grid cell. The updated Figure 4 is listed below (the figure indexes in this text are the same to these in original manuscript):

[Figure]

**Figure 4.** (a) A fine-scale DEM of Antarctica at a posting of 500 m derived from ICESat-2, which covers both the ice sheet and ice shelves with the southern limit of 88°S. (b) Map of the ICESat-2 DEM elevation uncertainty.

Second, in order to provide a robust and reasonable comparison between ICESat-2 DEM and other DEMs, a common OIB data in areas of low elevation change from 2009 to 2019 are used in the revised manuscript. As the same evaluation data are used, all the DEMs can be compared validly. These OIB data are located in the ice sheet interior, as shown in the figure below (Figure 1b). The CryoSay-2 Low Rate Mode (which was designed for flat ice sheet interior measurements) mask is used to extract the regions of low elevation change. CryoSat Geographical Mode Mask (v 4.0, updated in 19 August - 26 August 2019) at https://earth.esa.int/eogateway/news/cryosat-geographical-mode-mask-4-0-released is used here. The averaged elevation change rate in the used

OIB data locations is about -0.0074±0.0821 m/yr from 2003 to 2019, according to elevation change rate data from Smith et al. (2020). Hence, we assume that in these areas the effect of the elevation change on the DEM evaluation can be ignored. It should be noted that, when evaluating ICESat-2 DEM individually, OIB data in areas of low elevation change from 2009 to 2017 and OIB data in all Antarctica from 2018 to 2019 (Figure 1a) are used for a comprehensive evaluation. In addition, published GPS transects (Schröder et al., 2017) are also used here for the DEM comparison (Figure 1c).

[Figure]

Elevation (m)

0    500   1000   1500   2000   2500   3000   3500   4000

**Figure 1.** (a) Map of the OIB airborne data in October and November 2018 and October and November 2019 in Antarctica. (b) Map of the OIB airborne data from 2009 to 2019 in Antarctic ice sheet interior. (c) Map of the GPS transects from 2001 to 2015 in Antarctica. The dashed lines in (b) and (c) show the boundary of region where we assume to have low elevation change, it is the mode mask boundary of CryoSat-2 Low Rate Mode data in Antarctica, this data mode was designed for flat ice sheet interior measurements.

In addition, in the revised manuscript we resample all the DEMs to the OIB data locations and calculate the difference and statistics to perform the evaluation. The original approach (calculating a median OIB elevation for each grid cell) will certainly influence the evaluation results as the DEMs have different pixel spacing. Also as pointed out by the reviewers, OIB is the reference elevation and cannot be replaced by the median values. The new approach is also applied for DEM evaluation using GPS data. The related statistics have changed, but the same conclusions are found comparing to original statistics. The updated Tables 3, 5 and 6 are listed in below:

**Table 3.** Comparisons between the ICESat-2 DEM and OIB airborne elevation measurements (including data in areas of low elevation change from 2009 to 2017 and data in all Antarctica from 2018 to 2019) in observed and interpolated areas for individual regions (i.e., the ice sheet and ice shelves).

|  | Region | MeD (m) | MeAD (m) | SD (m) | RMSD (m) | Number of used OIB measurement points |
|---|---|---|---|---|---|---|
| Observed | Ice sheet | 0.08 | 1.18 | 12.75 | 12.75 | 3589087 |
|  | Ice shelves | 0.77 | 2.60 | 15.26 | 15.27 | 191754 |
|  | Total | 0.09 | 1.23 | 12.89 | 12.89 | 3780841 |
| Interpolated | Ice sheet | -0.40 | 2.50 | 20.68 | 20.73 | 1237416 |
|  | Ice shelves | 0.36 | 3.23 | 24.61 | 24.65 | 185613 |

|  |  |  |  |  |  |  |
|---|---|---|---|---|---|---|
|  | Total | -0.33 | 2.58 | 21.25 | 21.28 | 1423029 |
| Overall | Ice sheet | 0.01 | 1.41 | 15.20 | 15.20 | 4826503 |
|  | Ice shelves | 0.59 | 2.88 | 20.40 | 20.43 | 377367 |
|  | Total | 0.03 | 1.49 | 15.64 | 15.64 | 5203870 |

**Table 5.** Comparisons between the ICESat-2 DEM, ICESat DEM, ICESat/ERS-1 DEM, Helm CryoSat-2 DEM, Slater CryoSat-2 DEM, REMA DEM, TanDEM PolarDEM and OIB airborne elevation measurements in areas of low elevation change from 2009 to 2019.

|  | MeD (m) | MeAD (m) | SD (m) | RMSD (m) | Number of used OIB measurement points |
|---|---|---|---|---|---|
| ICESat-2 DEM | 0.10 | 0.98 | 5.36 | 5.38 |  |
| ICESat DEM | -2.61 | 6.35 | 19.90 | 20.43 |  |
| ICESat/ERS-1 DEM | -0.15 | 1.84 | 11.53 | 11.54 |  |
| Helm CryoSat-2 DEM | 0.65 | 2.68 | 24.97 | 25.02 | 1965309 |
| Slater CryoSat-2 DEM | 1.22 | 2.87 | 23.85 | 24.14 |  |
| REMA DEM | -0.16 | 0.53 | 1.75 | 1.76 |  |
| TanDEM PolarDEM | -2.84 | 2.94 | 2.76 | 3.90 |  |

**Table 6.** Comparisons between the ICESat-2 DEM, ICESat DEM, ICESat/ERS-1 DEM, Helm CryoSat-2 DEM, Slater CryoSat-2 DEM, REMA DEM, TanDEM PolarDEM and GPS elevation data in areas of low elevation change from 2001 to 2015.

|  | MeD (m) | MeAD (m) | SD (m) | RMSD (m) | Number of used OIB measurement points |
|---|---|---|---|---|---|
| ICESat-2 DEM | -0.03 | 0.41 | 1.17 | 1.17 |  |
| ICESat DEM | -1.91 | 2.89 | 5.21 | 5.97 |  |
| ICESat/ERS-1 DEM | -0.74 | 0.84 | 1.39 | 1.61 |  |
| Helm CryoSat-2 DEM | 0.07 | 0.67 | 1.67 | 1.71 | 488963 |
| Slater CryoSat-2 DEM | 0.00 | 0.46 | 1.65 | 1.66 |  |
| REMA DEM | 0.03 | 0.26 | 0.57 | 0.57 |  |
| TanDEM PolarDEM | -4.62 | 4.62 | 1.33 | 4.72 |  |

References:

Smith B, Fricker H A, Gardner A S, et al. Pervasive ice sheet mass loss reflects competing ocean and atmosphere processes. Science, 2020, 368(6496): 1239-1242.

Schröder L, Richter A, Fedorov D V, et al. Validation of satellite altimetry by kinematic GNSS in central East Antarctica. The Cryosphere, 2017, 11(3): 1111-1130.

General comments

Shen et al. describe a new DEM of Antarctica generated from 1 year of IceSat-2 data, using a model fit and blended resolution approach similar to Slater et al., 2018. The DEM is compared to airborne laser altimeter data to assess its accuracy, and compared to other DEMs derived from both radar and laser altimetry, radar interferometry and stereo-photogrammetry. The paper is generally well written

and structured, although the readability of most figures could be improved, in terms of resolution and the choice of colour blind friendly colour scales. A new DEM exploiting the high accuracy and density of IceSat-2 data is a welcome product and is worthy of publication. However, I have some concerns relating to the description of the DEM resolution, model fit, and the comparison to both OIB data and other DEMs which I would appreciate if the authors could address.

Thank you very much for your feedback and advice, these suggestions have significantly improved the text and figures. We have revised the manuscript according to all your comments, the details can be found in below.

DEM resolution and posting – the authors claim the modal resolution of the DEM is 250 m, but to me this does not seem correct. While the DEM is posted at 250 m, the most commonly used model fit is 1 km, and the majority of the DEM comprises of 500 m and 1 km model fits resampled to 250 m.

Agree and accept. As we mentioned in the very beginning, we produce a new ICESat-2 DEM at 500 m posting, the 500m DEM is refilled with estimated and interloped DEMs at 1 km posting to give a more reasonable and robust elevation estimate. The incorrect statement has been deleted.

Model fit method – I have some concerns with the authors choice of using the model fit method, which I would appreciate if they could address:

A linear component in time is more appropriate for longer time series, not one year of data where this parameter will be poorly constrained. How effective is this model at separating temporal elevation changes with just one year of data?

The map of elevation change rate in this study is shown in below, we also provide the estimation result from 2003 to 2019 in Smith et al. (2019) for a comparison. Overall, similar elevation change patterns can be found between the two figures. For example, larger elevation decreases can be found in the margin of West Antarctica, obvious elevation increases can be found in the interior of West Antarctica (red cycles in the figure). Hence, the elevation change pattern based on one-year ICESat-2 data is reasonable, which indicates that one year of data can give a reliable elevation change map and the elevation estimation is thus reliable. This may due to the much higher measurements density and accuracy of ICESat-2 than previous altimeters. In addition, ICESat-2 DEM has a higher accuracy than other altimeter-derived DEMs by comparing to both airborne and GPS data, which also proves the feasibility of the data and method.

[Figure]

**Figure.** Map of elevation change rate in Antarctica derived from one year of ICESat-2 data in this study (left) and map of elevation change rate in Antarctica from 2003 to 2019 in Smith et al. (2019) (right).

Reference:

Smith B, Fricker H A, Gardner A S, et al. Pervasive ice sheet mass loss reflects competing ocean and atmosphere processes. Science, 2020, 368(6496): 1239-1242.

Fitting a model to IceSat-2 data, which is both high accuracy and high density, to me this is degrading the spatial sampling provided by this dataset, which should resolve finer scale features not observed by e.g. a larger radar footprint. The paper would benefit from the author's adding a bit of text to justify why this approach is best for IceSat-2.

Here we add a statement about the choice of model fitting method:

A model fitting method used here is to separate the various contributions to the estimated elevations within each grid cell (Flament and Remy, 2012; McMillan et al., 2014), including local surface terrain and elevation change. This function is fitted in each grid cell by using an iterative least-squares fit to all the elevation measurements to minimize the impact of outliers. A quality control criterion is also used to reduce the effect of any poor fit. This method suits ICESat-2 orbit cycle, which samples dense ground tracks comparing to previous satellite radar altimeters, more measurement points are included in the grid cell and the estimated elevations are more robust. The resolutions of grid cells (i.e., 500 m and 1 km) are appropriate for the used ICESat-2 data in this study. First, most elevations (72%) can be directly estimated based on this method. Second, it is possible for a quadratic form to model the topography at these scales and smaller elevation residuals can be found than using a simple linear fit (Flament and Remy, 2012).

In addition, model fitting method can provide the estimation of elevation change rate, and the estimate agrees well with accurate elevation change estimations from crossover-point method

(Moholdt et al., 2010), which provides an addition reference for the research of ice dynamics and mass balance.

References:

Moholdt G, Nuth C, Hagen J O, et al. Recent elevation changes of Svalbard glaciers derived from ICESat laser altimetry. Remote Sensing of Environment, 2010, 114(11): 2756-2767.

Flament T, Rémy F. Dynamic thinning of Antarctic glaciers from along-track repeat radar altimetry. Journal of Glaciology, 2012, 58(211): 830-840.

McMillan M, Shepherd A, Sundal A, et al. Increased ice losses from Antarctica detected by CryoSat-2. Geophysical Research Letters, 2014, 41(11): 3899-3905.

Slater T, Shepherd A, McMillan M, et al. A new digital elevation model of Antarctica derived from CryoSat-2 altimetry. The Cryosphere, 2018, 12(4): 1551-1562.

Comparison to OIB – the authors restrict the OIB comparison of their DEM due to temporal differences between the two datasets. This severely limits the amount of OIB data available for comparison. To my mind, temporal differences in elevation between the two datasets will only be worth considering in regions where elevation trends are high due to either ice dynamics or surface mass balance anomalies - why haven't the authors used OIB data from other time periods in the interior of the ice sheet for example, where the surface height will be stable over time? This would allow more OIB data to be used and the DEM accuracy to be more robustly assessed.

Agree and accept. This has been responded in the very beginning.

Comparison to other DEMs – because the authors have not used a common OIB dataset to compare against the other DEMs, this limits their ability to claim their DEM is the most accurate (please note I am not doubting this is the case). Using a common dataset, or adjusting for temporal changes in elevation would allow for a more robust comparison.

Agree and accept. This has been responded in the very beginning.

Specific comments

L10 – I guess this should be 'thirty years', not 'thirty decades'?

Yes and corrected.

L51 – it is more that spatial and temporal variations in Ku band penetration depth are difficult to account for; I'd suggest re-wording this sentence to better reflect that

Agree and accept. We have revised this sentence to avoid the misleading information:
'Although the radar penetration depth of the Ku-band into snowpack can be corrected for either empirically or theoretically using a waveform fitting approach (Davis, 1996; Davis, 1997), the spatial and temporal variations of penetration depth are still difficult to account. As multi-temporal and large-scale satellite altimeter data are usually used, this includes some uncertainties in the

elevation estimation.'.

The original sentence is also listed below:

References:
Davis C H. Temporal change in the extinction coefficient of snow on the Greenland ice sheet from an analysis of Seasat and Geosat altimeter data. IEEE transactions on geoscience and remote sensing, 1996, 34(5): 1066-1073.
Davis C H. A robust threshold retracking algorithm for measuring ice-sheet surface elevation change from satellite radar altimeters. IEEE Transactions on Geoscience and Remote Sensing, 1997, 35(4): 974-979.

L74 – how are 'good quality' data defined – or on which criteria are poor quality data thrown out?

The surface signal confidence metric (i.e., atl06_quality_summary) from ICESat-2 data is used as the data filter criterion. Only ICESat-2 data with atl06_quality_summary values of zero are used. The detailed introduction to atl06_quality_summary can be referred to Smith et al. (2019).

Here we reworded this sentence to give a clear statement:
'… here, only ATL06 data with good quality (according to the surface signal confidence metric from ATL06 data, i.e, those for which atl06_quality_summary equals zero) are used to generate the DEM.'.

The original sentence is:

Reference:
Smith B, Fricker H A, Holschuh N, et al. Land ice height-retrieval algorithm for NASA's ICESat-2 photon-counting laser altimeter. Remote Sensing of Environment, 2019, 233: 111352.

L87 – Not sure what the authors are referring to here by 'seasonal elevation changes'? Seasonal elevation changes in Antarctica are only really specific to the Peninsula.

It should be sub-annual changes, all related statements have been revised.

Fig 1 – suggest making the axis labels larger as they're difficult to make out

The resolution of all figures has been improved. As Figs. 1b, c and d do not provide useful information in the text, hence we have removed them in the revised manuscript (also as suggested by another reviewer).

Table 1 – this table is misleading as all the altimeter derived DEMs do have timestamps. Helm et al (2014) is derived from one cycle of data and Bamber et al (2009) correct for elevation changes between acquisition period of the two datasets. Instead of saying 'unclear', it would be more appropriate to state the acquisition periods of the datasets used. I'm also unclear on what is meant by 'Pan-Antarctica', as Bamber et al (2009) includes the ice shelves also?

Agree and accept, all DEMs have specific time stamps, but the temporal resolutions are different. In order to avoid the misleading information in original Table 1, we delete the column [time stamp] as the column [Time span of applied source data] has provided the information for DEM time stamps.

Thank you for your correction, the column [Coverage] has be removed. All the information in Table 1 have been checked, the updated Table 1 is also shown below:

**Table 1.** Detailed introductions to six previously published Antarctic DEMs, including the source data, time span of the source data, spatial posting/resolution.

| DEM | Source data | Time span of applied source data | Spatial posting/resolution |
| --- | --- | --- | --- |
| ICESat DEM | ICESat | February 2003 to June 2005 | 500 m |
| ICESat/ERS-1 DEM | ICESat, ERS-1 | 1994-1995, 2003-2008 | 1 km |
| Slater CryoSat-2 DEM | CryoSat-2 | July 2010 to July 2016 | 1 km |
| Helm CryoSat-2 DEM | CryoSat-2 | A full 369-day-long cycle starting January 2012 | 1 km |
| REMA DEM | GeoEye-1, WorldView-1/2/3 | As of July 2017 | Variable resolutions, 2 and 8 m |
| TanDEM-X PolarDEM | TerraSAR-X, TanDEM-X | April to November 2013, April to October 2014, mid-2014, July 2016 to September 2017 | 90 m |

L128 'gaps' not 'gasps'

Done.

L141/Figure 2 – why have the authors chosen to show data density at 1 km, and not the posting of the DEM (250 m)?

Accept. Here we show the data density at 500 m for the new DEM (500 m posting).

[Figure]

**Figure 2.** Map of the numbers of valid ICESat-2 measurement points in each 500 m grid cell. The numbers of ICESat-2 measurement points in 1 km grid cells are resampled to the resolution of 500 m.

L163 – I'm confused by the author's claim that the modal resolution of the DEM is 250 m in the abstract – if most spatial coverage is provided by 1 km model fits then is that not the modal resolution?

We have deleted this incorrect statement.

L171 – I would suggest rewording as I feel this sentence is misleading – the DEM is posted at a resolution of 250 m, but the resolution is not 250 m as the most commonly used model fit is 1 km. This should be addressed elsewhere in the text (particularly the abstract) to make this clear to the reader.

Accept. We have reworded this sentence:
'Although two resolutions are applied, 1 km and interpolated elevations are all resampled to a posting of 500 m to provide a consistent DEM dataset; hence, the final ICESat-2 DEM is posted at a resolution of 500 m.'.

The original sentence is:

The related statements in the text have also been revised.

Fig 3 – I'm surprised to see such large differences (up to ~ 300 m) between the three different resolutions? This could mean that the model fit is not working as intended; if the authors could investigate further into e.g. the spatial distribution of these differences that may help understand what's happening

In the Fig. 3, elevations at different latitudes are presented by the region-averaged values. As shown in the Fig. 2 (in below), the coverages of DEMs at three resolutions are different, the observed elevation grid cells at the same region are not completely overlapped, DEM at coarser resolution has more observed grid cells than DEM at fine resolution at the same region, hence the region-averaged elevation values have large differences.

[Figure]

**Figure 2.** Map of the observed grid cells of DEMs at the spatial resolution of 250 m (a), 500 m (b) and 1 km (c). The observed grid cells are colored in blue, the overall coverage of each DEM in Antarctica is also presented beside.

In the revised manuscript, we compare the elevations at two resolutions (i.e., 500 m and 1 km, as we mentioned in the very beginning only two resolutions are used in the revised manuscript) in the overlapped regions in Antarctica. The map and histogram of elevation difference between 1km DEM and 500m DEM are shown in Figs. 3b and 3c. An averaged elevation difference of 0.04 ± 2.93 m is found, which is quite small comparing to the estimated elevations. This elevation difference is acceptable and it is valid to combine elevations at these two resolutions.

[Figure]

**Figure 3.** (a) Spatial coverages of observed grid cells in the five latitude ranges when two spatial resolutions, i.e., 500 m (blue) and 1 km (red), are applied. (b) Map of the elevation difference of DEMs at the resolutions of 1 km and 500 m. (c) Histograms of the elevation difference of DEMs at the resolutions of 1 km and 500 m, the average and standard deviation values are also presented.

Fig 4 – I'm not sure if the colour scale is playing tricks on me but it seems that the uncertainty is larger for the much of the ice shelves than it is for the ice sheet margins? Could the authors please

explain why this is the case? The ice shelves are flat so the uncertainty should be lower here I think?

Most ice shelves are coloured in green/yellow while the ice sheet margins are coloured in yellow/red, hence uncertainties in ice sheet margins are generally larger than these in ice shelves. This has been confirmed by checking the original data, an averaged elevation uncertainty of 0.58 ± 22.67 m is found for ice shelves, while for ice sheet margins (within the CryoSat-2 Low Rate Mode mask) the number is 0.93 ± 21.78 m. In addition, as the finer resolutions and one-year data used in this study, there are some observed elevation gaps in low latitudes (e.g., ice shelves), these elevations are estimated by using interpolation method. The uncertainty values of interpolated elevations are usually larger than these from observed elevation. This explains why similar uncertainty values can be found in some areas of ice shelves and ice sheet margins.

Fig 7 – I find this figure hard to read, improved resolution and particularly the colour scale used in panel c would improve the readability of this figure

The resolutions of all figures in the manuscript have been improved. The colors of 7a and 7c are the same in the updated figure.

[Figure]

Figure 7. (a) Coverages of observed grid cells at 500 m and 1 km and interpolated grid cells in 27 drainage basins of ice sheets (Zwally et al., 2012) and ice shelves. The boundaries and basin index (ID) of 27 ice sheet drainage basins (Numbers 1 to 27) and ice shelves (Number 28) are shown in (b). The coverages of observed (at two spatial resolutions) and interpolated grid cells in the Antarctic Peninsula, West Antarctica, East Antarctica and ice shelves are also shown in (b). (c) Map of the selected grid cell resolution for deriving the ICESat-2 DEM in all grid cells at a spatial resolution of 500 m. Elevation values derived from 1 km and interpolation (i.e., 1 km) are resampled to a resolution of 500 m.

L246 – remove 'obviously'

Done.

Fig 8 – Suggest using a colour blind friendly colour scale here

Accept. Red-white-blue like in Figure 9 has been applied in this figure.

[Figure]

Figure 8. (a) Map of the difference between the ICESat-2 DEM and OIB airborne elevation measurements. Detailed maps of the ice sheet interior, Pine Island Glacier region and ice sheet margin in East Antarctica are shown in (b), (c) and (d), and their locations are also shown in (a) by red rectangular boxes. The background is the shaded relief map of Antarctica derived from the ICESat-2 DEM.

Table 5 – I realise the authors have done this because the DEMs have different timestamps, but this is not a fair comparison as different subsets of OIB data are used for each DEM, so it's not possible to compare between the two. As mentioned previously, I don't see the need for the authors to restrict OIB data in time in areas of low elevation change, so that could be a way to perform a more fair comparison. It may also be possible to e.g. correct for longer term elevation change between the two datasets using contemporaneous elevation trends.

Agree and accept. These comments have been responded in the very beginning.

Table 6 – I noticed the number of grid compared grid cells are different here – does this Table use different subsets of OIB data also?

The same OIB data were used in the original Table 6. As the different grid cell resolutions of listed DEMs, the final grid cells used for comparison are different. In the revised manuscript this table has been removed as we used a common OIB data and GPS data for comparisons. Now we resample all the DEMs into OIB/GPS locations, hence the number of the used OIB/GPS points are the same. Detailed responses can be found in the very beginning.

Best wishes,